# Variational Best-of-$N$ Alignment

**Afra Amini**   **Tim Vieira**   **Elliott Ash**   **Ryan Cotterell**
ETH Zürich
{afra.amini, ryan.cotterell}@inf.ethz.ch
tim.f.vieira@gmail.com  ashe@ethz.ch

## Abstract

Best-of-$N$ (Bo$N$) is a popular and effective algorithm for aligning language models to human preferences. The algorithm works as follows: at inference time, $N$ samples are drawn from the language model, and the sample with the highest reward, as judged by a reward model, is returned as the output. Despite its effectiveness, Bo$N$ is computationally expensive; it reduces sampling throughput by a factor of $N$. To make Bo$N$ more efficient at inference time, one strategy is to fine-tune the language model to mimic what Bo$N$ does during inference. To achieve this, we derive the distribution induced by the Bo$N$ algorithm. We then propose to fine-tune the language model to minimize backward KL divergence to the Bo$N$ distribution. Our approach is analogous to mean-field variational inference and, thus, we term it variational Bo$N$ (vBo$N$). To the extent this fine-tuning is successful and we end up with a good approximation, we have reduced the inference cost by a factor of $N$. Our experiments on controlled generation and summarization tasks show that Bo$N$ is the most effective alignment method, and our variational approximation to Bo$N$ achieves the closest performance to Bo$N$ and surpasses models fine-tuned using the standard KL-constrained RL objective. In the controlled generation task, vBo$N$ appears more frequently on the Pareto frontier of reward and KL divergence compared to other alignment methods. In the summarization task, vBo$N$ achieves high reward values across various sampling temperatures.

 https://github.com/rycolab/vbon

## 1 Introduction

Language models are pre-trained on large corpora to model a distribution over natural language text.[1] Beyond their initial pre-training, they are often additionally fine-tuned on domain-specific data through a process called **supervised fine-tuning (SFT)**. The goal of SFT is to enable the model to better perform various downstream tasks of interest. While the fine-tuned model, called the **reference model** in our paper, is indeed typically much better at performing the downstream task of interest, e.g., dialogue generation or summarization, it may still generate undesirable content, e.g., harmful or offensive text. To mitigate this issue, **aligning** the reference model to human preferences has become a fundamental step in the development of modern large language models (Meta, 2023; OpenAI, 2023; Gemini, 2024).

The degree to which text is aligned with human preferences is typically operationalized using a real-valued reward function. Rather than constructing a reward function by hand, it is typically estimated from a dataset of human preferences.[2] And, after estimation, we expect the reward function to return higher values for text that is more likely to be preferred by humans, and lower values for text that is more likely to be dispreferred. Then, given an estimated reward function, an alignment algorithm further alters the reference models in a manner such that it places the highest probability on the text that is high reward under the reward model *and* high probability under the reference model.

Alignment algorithms can be taxonomized into two groups: (i) alignment via fine-tuning, where we change the language model's parameters to achieve alignment (Christiano et al., 2017; Rafailov

---

[1]Many language models are also used to model text in non-natural languages, e.g., programming languages.

[2]In some cases, the reward model is not estimated from human preference data. It is either known, e.g., code-based execution scores, or given by a classifier, e.g., toxicity or sentiment classifiers.

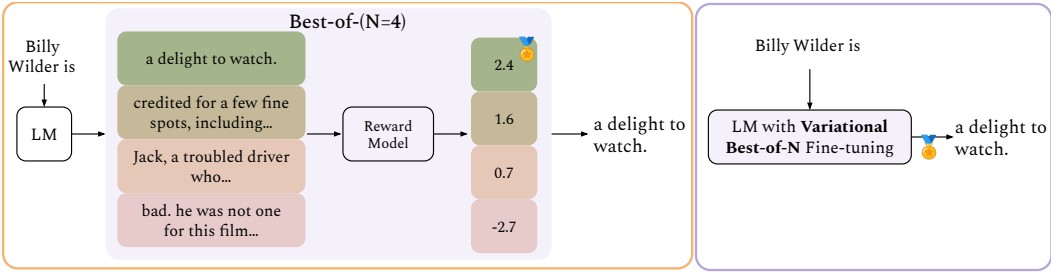

Figure 1: Best-of-$N$ (on the left) is an effective alignment-via-inference method: it draws $N$ samples from the language model, ranks them according to a reward model, and outputs the best sample. Variational Best-of-$N$ (on the right) approximates this process via fine-tuning. The goal is to ensure that sampling a single string from the fine-tuned model produces a result equivalent to applying Best-of-$N$. This approach allows us to achieve similar performance while increasing the throughput by a factor of $N$.

et al., 2023), and (ii) alignment via inference (Nakano et al., 2022; Mudgal et al., 2024). A common alignment-via-fine-tuning method is **reinforcement learning from human feedback** (**RLHF**; Christiano et al., 2017; Stiennon et al., 2020; Ouyang et al., 2022). RLHF typically consists of further fine-tuning the language model under a **KL-constrained RL objective**, which is made up of two terms: a term that encourages the model to maximize the reward, and a term that discourages high KL divergence between the language model and the reference model. This objective is often maximized with an RL algorithm, e.g., proximal policy optimization (PPO; Schulman et al., 2017). A common alignment-via-inference method is the Best-of-$N$ (Bo$N$; Stiennon et al., 2020) algorithm. As such, it does *not* require any fine-tuning of the language model. The algorithm is straightforward: One draws $N$ samples from the reference model and returns the text that achieves the highest reward among those $N$ samples. The Bo$N$ algorithm has also been effectively applied in controlled decoding (Yang & Klein, 2021; Mudgal et al., 2024) and to generate a dataset for supervised fine-tuning (Meta, 2023).

Despite its simplicity, Bo$N$ has proven incredibly practical in generating high-reward text that still has a high probability under the reference model. Theoretically, Yang et al. (2024) prove that under some simplifying assumptions, the Bo$N$ distribution is asymptotically equivalent to the optimal distribution under the KL-constrained RL objective. Empirically, it has been repeatedly shown (Gao et al., 2023; Rafailov et al., 2023; Mudgal et al., 2024) that Bo$N$ often appears on the frontier of reward and KL curves, surpassing the performance of models fine-tuned with RLHF. However, the main factor preventing Bo$N$ from replacing fine-tuning methods for alignment is its significant computational overhead during inference. Even when sampling is done in parallel, Bo$N$ decreases the text generation throughput by a factor of $N$. This drawback limits its practicality for generating text from large language models.

To speed up Bo$N$, we devise a scheme to convert it into an alignment-via-fine-tuning algorithm rather than an alignment-via-inference algorithm. To this end, we first formally derive the probability distribution induced by the Bo$N$ algorithm. Then we approximate this distribution by minimizing the reverse KL divergence between the language model and the Bo$N$ distribution. This leads to an optimization objective that we refer to as the vBo$N$ objective. By analyzing a lower bound of this objective, we find that it behaves similarly to the KL-regularization objective in the limit, i.e., $N \to 1$ or $N \to \infty$. Importantly, the vBo$N$ objective has a unique and useful property: it is insensitive to applying any monotonically increasing function to the reward values. This distinctive feature, along with the empirical success of the Bo$N$ algorithm, suggests that the vBo$N$ objective is a promising and interesting objective to explore. Finally, we fine-tune the language model using PPO to optimize the vBo$N$ objective. Our scheme, depicted in Fig. 1, allows us to achieve performance close to that of the Bo$N$ algorithm while increasing the inference throughput by a factor of $N$.

We experiment with vBo$N$ on controlled generation and summarization tasks, comparing its performance to models fine-tuned using the KL-constrained RL objective. For controlled generation, our results indicate that models fine-tuned with the vBo$N$ objective are more likely to fall on the Pareto frontier of the reward vs. KL curve compared to other fine-tuning-based alignment methods. This suggests that vBo$N$ achieves a better balance between maximizing reward and maintaining

proximity to the reference model. On a summarization task, fine-tuning with vBoN yields higher reward values and greater win rates on average than models fine-tuned with the KL-constrained RL objective, further demonstrating its effectiveness.

## 2    BACKGROUND: REINFORCEMENT LEARNING FROM HUMAN FEEDBACK

Let $\Sigma$ be an **alphabet**, a finite, non-empty set of symbols.[3] The elements of $\Sigma$ may be characters, tokens, or words; the choice lies with the modeler. A **string** is a finite sequence of symbols drawn from $\Sigma$. A **language model** is a distribution over strings $\boldsymbol{y} \in \Sigma^*$, where $\Sigma^*$ is the set of all strings over the alphabet $\Sigma$. In this paper, we consider language models, e.g., those based on neural networks, that are parameterized by a real vector $\boldsymbol{\theta} \in \boldsymbol{\Theta}$, denoted as $\pi_{\boldsymbol{\theta}}$. Furthermore, we restrict ourselves to language models that are differentiable functions of $\boldsymbol{\theta}$. In conditional generation tasks, e.g., summarization or dialogue generation, it is desirable to prompt the language model with a string $\boldsymbol{x} \in \Sigma^*$. Consequently, we consider prompted language models, i.e., those that give a conditional distribution over response strings $\boldsymbol{y}$, given a prompt string $\boldsymbol{x}$, as $\pi_{\boldsymbol{\theta}}(\boldsymbol{y} \mid \boldsymbol{x})$. However, for notational convenience, we will drop the explicit conditioning on the prompt $\boldsymbol{x}$ and simply write $\pi_{\boldsymbol{\theta}}(\boldsymbol{y})$.

Algorithms for RLHF fine-tune the language model to increase the expected reward of the strings sampled from it while not diverging too far from the reference model. RLHF consists of three steps. First, the language model is fine-tuned on a task-specific dataset using the maximum-likelihood objective. Recall we term the language model after this step the reference model and show that with $\pi_{\mathrm{ref}}$. Next, a **reward model** $r \colon \Sigma^* \to \mathbb{R}$ is trained to capture human preferences; the reward of a string is high if it is preferred by humans.[4] Finally, the reference model is fine-tuned to maximize the KL-constrained RL objective,

$$\mathcal{J}^{\mathrm{RL}}(\boldsymbol{\theta}) = \underset{\boldsymbol{y} \sim \pi_{\boldsymbol{\theta}}}{\mathbb{E}}\Big[r(\boldsymbol{y})\Big] - \beta\, D_{\mathrm{KL}}\big(\pi_{\boldsymbol{\theta}} \,\|\, \pi_{\mathrm{ref}}\big), \tag{1}$$

where $D_{\mathrm{KL}}(\cdot)$ is the KL divergence between two distributions, modulated by a hyperparameter $\beta$. This objective encourages the model to assign greater probability mass to high-reward outputs while simultaneously penalizing excessive divergence from the reference model. Levine (2018) shows that the policy that maximizes[5] this objective (Eq. (1)) is

$$\pi_{\boldsymbol{\theta}}^{\star}(\boldsymbol{y}) = \frac{1}{Z}\, \pi_{\mathrm{ref}}(\boldsymbol{y}) \exp\Big(\frac{1}{\beta} r(\boldsymbol{y})\Big), \quad Z = \sum_{\boldsymbol{y} \in \Sigma^*} \pi_{\mathrm{ref}}(\boldsymbol{y}) \exp\Big(\frac{1}{\beta} r(\boldsymbol{y})\Big). \tag{2}$$

In simple terms, $\pi_{\boldsymbol{\theta}}^{\star}$ is the reference model reweighted by the exponentiated reward values and normalized by the partition function $Z$. However, direct sampling from $\pi_{\boldsymbol{\theta}}^{\star}$ is not feasible, as computing $Z$ requires evaluating an infinite sum, making it intractable. However, a heuristic approach to sampling from $\pi_{\boldsymbol{\theta}}^{\star}$ would be to sample many strings from $\pi_{\mathrm{ref}}$ and only keep those that have high rewards. Indeed, this heuristic is the motivation behind the BoN algorithm.

## 3    DERIVING THE BEST-OF-$N$ OBJECTIVE

Best-of-$N$ is a simple alignment-via-inference algorithm. The algorithm works as follows. Let $Y_N = \{\boldsymbol{y}^{(n)}\}_{n=1}^N$ be the multi-set containing $N$ i.i.d. samples from $\pi_{\mathrm{ref}}$. Then, BoN returns $\boldsymbol{y}^{\star}$, where[6]

$$\boldsymbol{y}^{\star} = \underset{\boldsymbol{y}^{(n)} \in Y_N}{\operatorname{argmax}}\, r(\boldsymbol{y}^{(n)}). \tag{3}$$

We present the probability distribution induced by BoN with $\pi_{\mathrm{bon}}$. Notably, $\pi_{\mathrm{bon}}$ is *not* the optimal distribution under Eq. (1), the KL-constrained RL objective.[7] Despite this, the BoN algorithm often

---

[3]Please refer to Tab. 3 for a summary of notations used throughout the paper.

[4]For example, in a summarization task, a preference dataset consists of a document, two candidate summaries for that document, and a label indicating which summary is preferred by humans. The reward model is trained on this dataset to maximize the likelihood of correctly predicting human preference.

[5]This formulation implicitly assumes that there exists a $\boldsymbol{\theta} \in \boldsymbol{\Theta}$ that achieves the unconstrained maximum.

[6]We assume that the $\operatorname{argmax}$ is unique, or ties are broken in a well-defined manner.

[7]Under simplifying assumptions is $\pi_{\mathrm{bon}}$ asymptotically (in string length) equivalent to $\pi_{\boldsymbol{\theta}}^{\star}$ (Yang et al., 2024).

performs well—even in comparison to RLHF-based methods. This naturally raises the question: under what optimization objective is $\pi_{\mathrm{bon}}$ the optimal distribution? To answer this question, we first compute the probability of strings under $\pi_{\mathrm{bon}}$.

**Proposition 1.** *Suppose* $r\colon \Sigma^* \to \mathbb{R}$ *is a one-to-one mapping. Then, the probability of a string* $\boldsymbol{y}$ *under* $\pi_{bon}$ *is given by*

$$\pi_{bon}(\boldsymbol{y}) = \sum_{i=1}^{N} \binom{N}{i} \mathrm{F}\big(r(\boldsymbol{y})\big)^{N-i} \pi_{\mathrm{ref}}(\boldsymbol{y})^i, \qquad \mathrm{F}\big(r(\boldsymbol{y})\big) \stackrel{\text{def}}{=} \mathop{\mathbb{P}}_{\boldsymbol{y}' \sim \pi_{\mathrm{ref}}} \big(r(\boldsymbol{y}') < r(\boldsymbol{y})\big). \qquad (4)$$

*Proof.* See App. B. ∎

F can be understood as the strict cumulative density function of reward values under $\pi_{\mathrm{ref}}$. In other words, $\mathrm{F}\big(r(\boldsymbol{y})\big)$ represents the probability that a random sample drawn from $\pi_{\mathrm{ref}}$ has a reward value less than $r(\boldsymbol{y})$. We now describe how to fine-tune the language model to approximate $\pi_{\mathrm{bon}}$. Similar to variational inference, we minimize the reverse KL divergence between $\pi_{\boldsymbol{\theta}}$ and $\pi_{\mathrm{bon}}$. Concretely,

$$\mathcal{J}^{\mathrm{vBON}}(\boldsymbol{\theta}) = -D_{\mathrm{KL}}\big(\pi_{\boldsymbol{\theta}} \,||\, \pi_{\mathrm{bon}}\big) = \mathop{\mathbb{E}}_{\boldsymbol{y} \sim \pi_{\boldsymbol{\theta}}} \Big[ \log \pi_{\mathrm{bon}}(\boldsymbol{y}) - \log \pi_{\boldsymbol{\theta}}(\boldsymbol{y}) \Big] \qquad (5\text{a})$$

$$= \mathop{\mathbb{E}}_{\boldsymbol{y} \sim \pi_{\boldsymbol{\theta}}} \Big[ \log \pi_{\mathrm{bon}}(\boldsymbol{y}) \Big] + \mathrm{H}\big(\pi_{\boldsymbol{\theta}}\big) \qquad (5\text{b})$$

$$= \mathop{\mathbb{E}}_{\boldsymbol{y} \sim \pi_{\boldsymbol{\theta}}} \Big[ \log \sum_{i=1}^{N} \binom{N}{i} \mathrm{F}\big(r(\boldsymbol{y})\big)^{N-i} \pi_{\mathrm{ref}}(\boldsymbol{y})^i \Big] + \mathrm{H}\big(\pi_{\boldsymbol{\theta}}\big), \qquad (5\text{c})$$

where $\mathrm{H}(\cdot)$ is the entropy of a distribution. Thus, Eq. (5) offers an answer to the question of what objective BoN optimizes. Inspecting the objective further, we see that Eq. (5) is an entropy-regularized objective, where we use the probability of the string under the BoN distribution as the reward and discourage the model from having low entropy.

**Monotonically invariant.** An important property of the variational BoN objective is that it is invariant to applying a strictly monotonically increasing function to rewards. This is because the vBoN objective relies on reward values solely through F, which, as defined in Eq. (4), only depends on the ranking between the reward values and not their exact magnitude. This suggests that the vBoN objective may be less sensitive to outliers and the scale of rewards. This property is important as RL algorithms are notoriously sensitive to the scale of reward values (Henderson et al., 2018; Schaul et al., 2021).

**Approximating** $\log \mathrm{F}(\cdot)$**.** Maximizing Eq. (5) requires us to compute $\log \mathrm{F}(\cdot)$ for any $r(\boldsymbol{y})$. This, however, is computationally expensive, as we have to sum over the probabilities of all strings that have rewards less than $r(\boldsymbol{y})$. Fortunately, we can instead maximize a lower bound of Eq. (5) using a Monte Carlo estimator of $\mathrm{F}(\cdot)$. Concretely, we can write $\mathrm{F}(\cdot)$ as an expectation,

$$\mathrm{F}\big(r(\boldsymbol{y})\big) = \mathop{\mathbb{E}}_{\boldsymbol{y}' \sim \pi_{\mathrm{ref}}} \big[ \mathbb{1}\{r(\boldsymbol{y}') < r(\boldsymbol{y})\} \big]. \qquad (6)$$

We approximate $\mathrm{F}\big(r(\boldsymbol{y})\big)$ using $M$ i.i.d. samples from $\pi_{\mathrm{ref}}$, termed $\boldsymbol{y}'^{(1)}, \dots, \boldsymbol{y}'^{(M)} \stackrel{\text{i.i.d.}}{\sim} \pi_{\mathrm{ref}}$, using which we compute $\widehat{\mathrm{F}}\big(r(\boldsymbol{y})\big) \stackrel{\text{def}}{=} \frac{1}{M} \sum_{m=1}^{M} \mathbb{1}\{r(\boldsymbol{y}'^{(m)}) < r(\boldsymbol{y})\}$. We then take the log of this Monte Carlo estimator as a biased, but consistent estimator of $\log \mathrm{F}(\cdot)$ in Eq. (5).[8] In §5.1, we empirically assess the number of samples needed for $\log \widehat{\mathrm{F}}$ to accurately approximate $\log \mathrm{F}$.

---

[8]Using Jensen's inequality, we show biasedness. Concretely, note the following lower bound

$$\log \mathrm{F}\big(r(\boldsymbol{y})\big) = \log \mathop{\mathbb{E}}_{\boldsymbol{y}'^{(1)}, \dots, \boldsymbol{y}'^{(M)}} \left[ \frac{1}{M} \sum_{m=1}^{M} \mathbb{1}\{r(\boldsymbol{y}'^{(m)}) < r(\boldsymbol{y})\} \right] \qquad (7\text{a})$$

$$\geq \mathop{\mathbb{E}}_{\boldsymbol{y}'^{(1)}, \dots, \boldsymbol{y}'^{(M)}} \left[ \log \left( \frac{1}{M} \sum_{m=1}^{M} \mathbb{1}\{r(\boldsymbol{y}'^{(m)}) < r(\boldsymbol{y})\} \right) \right], \qquad (7\text{b})$$

where Jensen's inequality is applicable because $\log$ is concave. Consistency can be shown with an application of the delta method (§5.5.4; Casella & Berger, 2001).

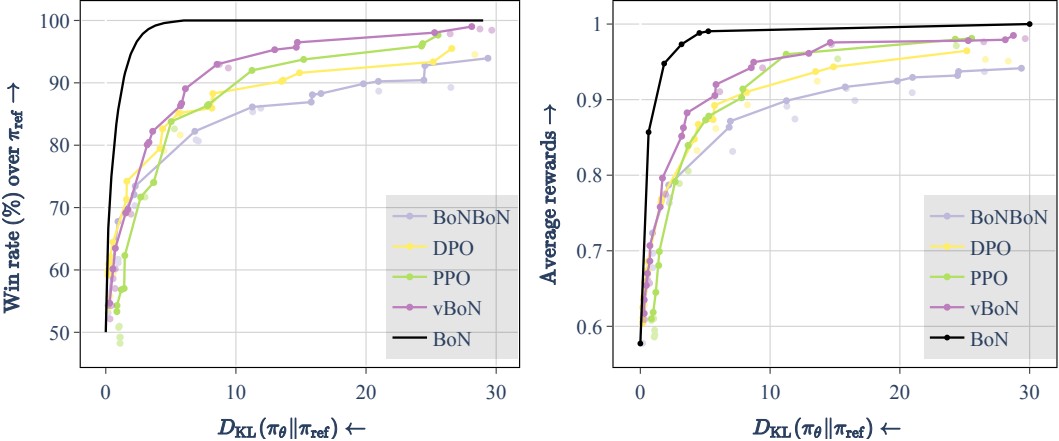

(a) 4% of points on Pareto front belong to BoNBoN, 4% to PPO, 42% to DPO, and 50% to vBo$N$.

(b) 7% of points on Pareto from belong to BoNBoN, 10% DPO, 33% PPO, and 50% vBo$N$.

Figure 2: Steering generated movie reviews towards positive sentiment. Points that are not on the Pareto front of each method have lower opacity. Bo$N$ is the most effective approach in achieving high win rates and high rewards while not diverging too far from the reference model. Our variational approximation to Bo$N$ gets closest to the performance of Bo$N$ compared to other fine-tuning methods, as reflected in the percentage of times it appears on the Pareto front.

## 4 COMPARING THE BO$N$ AND RL OBJECTIVES

To explore the connection between the vBo$N$ objective and the KL-regularized RL objective, we derive a lower bound for $\mathcal{J}^{\mathrm{vBON}}$. Through this lower bound, we hope to achieve a deeper insight into how the reward function is used in the variational Bo$N$ objective, and why this objective discourages high KL divergence from the reference model.

To derive such a lower bound, we substitute the Bo$N$ distribution in Eq. (4) into the vBo$N$ objective in Eq. (5). We then simplify this objective to arrive at the following theorem.

**Theorem 2.** *We have* $\mathcal{J}^{\mathrm{vBON}}(\boldsymbol{\theta}) \geq L(\boldsymbol{\theta})$, *where*

$$L(\boldsymbol{\theta}) \stackrel{\mathrm{def}}{=} (N-1) \mathop{\mathbb{E}}_{\boldsymbol{y} \sim \pi_{\boldsymbol{\theta}}} \left[ \log \mathrm{F}\big(r(\boldsymbol{y})\big) \right] - D_{\mathrm{KL}}\big(\pi_{\boldsymbol{\theta}} \,\|\, \pi_{\mathrm{ref}}\big). \tag{8}$$

*Proof.* See App. D. ∎

Empirically, we observe that models that are fine-tuned to maximize $L(\boldsymbol{\theta})$ perform competitively to the ones that are fine-tuned to maximize the vBo$N$ objective; see App. G for experimental results. Interestingly, if we compare Eq. (8) to the KL-constrained RL objective, Eq. (1), we see they have a very similar structure. We observe that $N$ (in the vBo$N$ objective) acts as a regularization parameter. As $N \to 1$, the optimal distribution gets closer to $\pi_{\mathrm{ref}}$, which has the same effect as $\beta \to \infty$ in Eq. (1). Furthermore, as $N \to \infty$, the optimal distribution only generates the string with the maximum rewards, which is equivalent to $\beta \to 0$ in Eq. (1). Importantly, in both limits, the optimal distribution under the KL-regularized RL objective and the vBo$N$ objective are equivalent.

The main difference between the KL-constrained RL objective Eq. (1) and the derived vBo$N$ lower bound Eq. (8) is in how the reward function is used. The KL-constrained RL objective aims to maximize the expected reward values, whereas vBo$N$ maximizes the cumulative probability that strings sampled from the aligned model, $\pi_{\boldsymbol{\theta}}$, achieve higher rewards compared to those sampled from $\pi_{\mathrm{ref}}$.

## 5 SENTIMENT CONTROL

We now employ the variational Bo$N$ objective, Eq. (5), to fine-tune language models. We perform an open-ended text generation task where the goal is to generate movie reviews with positive sentiment.

The reference model, $\pi_{\text{ref}}$, is GPT-IMDB[9], a GPT-2 (Radford et al., 2019) model fine-tuned on IMDB corpus (Maas et al., 2011). We use a binary sentiment classifier,[10] denoted as $p$, with two classes $\{\text{POS}, \text{NEG}\}$ as the reward model, and define $r(\boldsymbol{y}) \stackrel{\text{def}}{=} p(\text{POS} \mid \boldsymbol{y})$. Following Rafailov et al. (2023), we sample 5000 movie reviews from the training set of IMDB dataset and for each sample, we randomly choose a prefix length from $\{2, \dots, 8\}$ and take that prefix as the prompt. We further generate 512 prompts in the same way from the test set of IMDB that we use to evaluate our models.

We fine-tune the reference model with PPO using the vBo$N$ objective Eq. (5). Then, we compare the performance of the fine-tuned model (**vBo$N$**) to the exact Bo$N$ (**Bo$N$**), i.e., applying Bo$N$ at inference time.

We implement and compare the following existing methods for language model alignment:

- **Bo$N$-SFT:** Perhaps the most straightforward way to approximate Bo$N$ distribution is to fine-tune the model to maximize the likelihood of the samples taken with Bo$N$ algorithm. Unfortunately, we find that SFT is incapable of achieving a good trade-off between achieving high rewards and low KL divergence, see App. H (Fig. 7) for the experimental results.
- **PPO:** We use PPO to optimize the KL-constrained objective in Eq. (1). We use the default hyperparameters in trlx library (Havrilla et al., 2023) for fine-tuning with PPO.
- **DPO.** Direct preference optimization (DPO; Rafailov et al., 2023) is a popular alternative to RLHF that does not require training a reward model. Following DPO's experimental setup, we generate 6 reviews per prompt and use the resulting 12 pairwise comparisons per prompt to construct DPO's contrastive loss.[11]
- **BoNBoN:** Concurrent work (Gui et al., 2024) explores another approach to approximate Bo$N$ distribution. Assuming that the reference model distribution $\pi_{\text{ref}}$ is continuous, Gui et al. (Theorem 3; 2024) prove that the expected difference between the relative likelihood, i.e., $\frac{\pi_{\text{bon}}(\cdot)}{\pi_{\text{ref}}(\cdot)}$, of the Best-of-$N$ response and the Worst-of-$N$ response is $\frac{1}{2\beta} = \frac{1}{(N-1)\sum_{k=1}^{N-1} 1/k}$. They use this property to construct a loss function similar to that of IPO (Azar et al., 2023). Furthermore, they add another term to the loss function, which simply maximizes the likelihood of the Best-of-$N$ response. The final loss function is a convex combination of the IPO-like loss and the negative log-likelihood loss, regulated by a hyperparameter $\alpha$.[12]

We fine-tune models by varying the degree of regularization. For Bo$N$ approaches, that is achieved by varying $N$, and for DPO and PPO, we vary $\beta$.[13] Conveniently, $N$ in vBo$N$ is a hyperparameter, meaning that we do *not* need to generate more samples from $\pi_{\text{ref}}$ when we increase $N$. However, with Bo$N$ and BoNBoN methods, we need to increase the number of samples from the reference model as we increase $N$.

We generate movie reviews based on prompts from our test set using fine-tuned models and then measure three metrics: (i) KL divergence between the fine-tuned model and the reference model; (ii) win rate, defined as the percentage of times the fine-tuned model's generations receive higher rewards compared to the reference model's generations; and (iii) average rewards obtained by the fine-tuned model's sampled strings.

For the Bo$N$ method, we report the empirical upper bound of $\log N - \frac{N-1}{N}$ for KL divergence (Beirami et al., 2024; Mroueh, 2024) in our plots. Furthermore, the win rate of Bo$N$ over the reference model can be computed analytically and is equal to $\frac{N}{N+1}$.

We visualize the win rate vs. KL curves in Fig. 2a, and Fig. 2b the average rewards of generations under $\pi_{\boldsymbol{\theta}}$ vs. the KL divergence. As expected, Bo$N$ is the most effective approach; however, this comes at an extra inference cost that grows with $N$. We observe that among the fine-tuning methods, our variational approximation to Bo$N$ gets closest to the performance of Bo$N$, as it appears more

---

[9]Specifically, we use https://huggingface.co/lvwerra/gpt2-imdb.

[10]Specifically, we use https://huggingface.co/lvwerra/distilbert-imdb.

[11]One could argue that DPO has a slight advantage over other methods in this setup since it has seen 6 unique generations per prompt during training, while the others only have seen one (or 2 with BoNBoN). Nevertheless, we observe that vBo$N$ is more effective than DPO.

[12]Following the authors' recommendation, we set $\alpha$ so that both terms contribute equally to the final loss.

[13]See App. F for more details regarding the regularization hyperparameters.

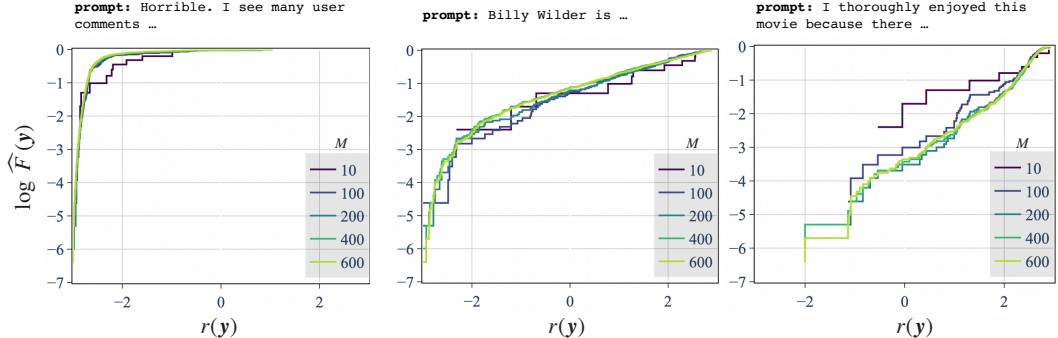

Figure 3: Estimates of $\log \mathrm{F}(\cdot)$ with increasing the number of Monte Carlo samples. We test an adversarial prompt (left plot), a neutral prompt (middle plot), and a prompt with a positive sentiment (right plot). Overall, we hardly see any difference between the estimates after taking 200 samples. For the adversarial prompt, the distribution of rewards is peaked, and we do not see any changes in our estimator after taking only 100 samples.

often on the Pareto front of the two curves compared to other methods. Notably, we observe that DPO performs better than PPO in terms of win rates but worse in terms of average rewards; this could be attributed to the contrastive nature of DPO's loss function.

## 5.1 Error in Estimating $\log \mathrm{F}(\cdot)$

We empirically quantify the error when estimating $\log \mathrm{F}(\cdot)$ with a finite number of i.i.d samples from $\pi_{\mathrm{ref}}$. To get a better intuition on the error of our estimators, in Fig. 3, we visualize the estimators for 3 different prompts: one adversarial prompt (left plot), where the prompt itself has a negative sentiment, one neutral prompt (middle plot), and one prompt with a positive sentiment (right plot). We vary the number of Monte Carlo samples from 10 to 600. We observe that for all the 3 prompts, the estimated CDF hardly changes after 200 samples. When using the adversarial prompt, the reward distribution is negatively peaked, and the estimated CDF does not change after taking only 100 samples.

We then quantify the change in the estimator by performing a two-sample Kolmogorov–Smirnov test (Hodges, 1958). This test measures the closeness of two empirical cumulative distribution functions. Concretely, the test statistic is

$$\sup_{\boldsymbol{y} \in \Sigma^*} \left| \widehat{\mathrm{F}}_{M_1}\big(r(\boldsymbol{y})\big) - \widehat{\mathrm{F}}_{M_2}\big(r(\boldsymbol{y})\big) \right|, \tag{9}$$

where $\widehat{\mathrm{F}}_{M_1}$ and $\widehat{\mathrm{F}}_{M_2}$ are estimated CDFs from $M_1$ and $M_2$ samples respectively. The statistics show the magnitude of the difference between the two empirical distributions of samples. The null hypothesis is that the two distributions are identical.

In Tab. 1, for each sample size $M$, we compare the estimated CDF with $M$ samples to the estimated CDF with 600 samples. If the two distributions are identical according to the test, we can reliably use the $M$ sample to estimate the CDF. We report the number of prompts (out of 5000 prompts) for which we reject the null hypothesis, meaning that the distributions are not identical. Furthermore, for those prompts, we report the average test statistics and $p$-values. In general, for very few prompts, the null hypothesis is rejected. Moreover, with 250 samples, the estimated CDFs are identical to the estimated CDF with 600 samples for all prompts.

Table 1: Measuring the estimation error with increasing the sample size. After 250 samples, the estimated CDF is unchanged for all the prompts.

| $M$ | Rejection rate | Test statistics | $p$-value |
|---|---|---|---|
| 5 | 6.14% | 0.63 | 0.02 |
| 20 | 4.02% | 0.33 | 0.03 |
| 100 | 1.14% | 0.17 | 0.02 |
| 200 | 0.06% | 0.12 | 0.02 |
| 250 | 0 | - | - |

## 5.2 Efficiency Analysis

We break down the efficiency analysis into 3 main parts: (i) the inference cost, (ii) the preference optimization cost, (iii) and the preprocessing cost.

**Inference cost.** As discussed earlier, vBo$N$ is an alignment-via-fine-tuning method, and along with other alignment-via-fine-tuning methods, it is $N$ times more efficient at inference compared to Bo$N$.

**Optimization cost.** We compare vBo$N$'s preference optimization cost to its closest alignment-via-fine-tuning counterpart, PPO. In the optimization loop, the main difference between PPO and vBo$N$ is that vBo$N$ requires computing the strict CDF function, F, using $M$ samples. Crucially, $N$ in vBo$N$ serves as a regularization hyperparameter, and increasing $N$ does *not* incur additional computation costs. To implement vBo$N$ efficiently, we precompute the F function before starting the optimization loop. This means the computational overhead is incurred only once, regardless of the number of optimization runs.[14] Since the F values are precomputed, we empirically observe that the time needed to run the vBo$N$ optimization loop *is the same as* running the PPO optimization loop, and the cost of evaluating F is negligible. Therefore, the main computational overhead in vBo$N$ comes from precomputing $\log \mathrm{F}(\cdot)$.

**Preprocessing cost.** Estimating $\log \mathrm{F}(\cdot)$ requires only forward passes through the LLM and reward model without the need to compute and store gradients. This makes the process highly parallelizable. Our experiments utilize a memory-efficient library for LLM inference (vLLM; Kwon et al., 2023), which allows us to perform these approximations efficiently.

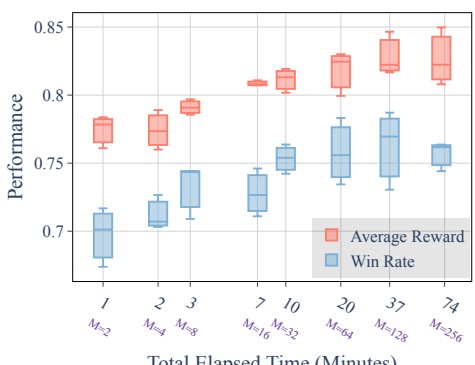

We examine the impact of increasing the computational cost of vBo$N$ by varying $M$, which directly affects the total elapsed time and downstream performance. For this analysis, we fix $N = 10$ and fine-tune the model using three random seeds. We report the average and standard deviation of reward values and win rates in Fig. 4 on a single A100-40GB GPU. Our results show that increasing $M$ generally improves the aligned model's rewards and win rates. Notably, even with $M = 32$ samples (taking only 10 minutes), the performance remains competitive with higher values of $M$. We hypothesize that the data efficiency of the simple Monte Carlo estimator can be improved by taking into account the similarity

Figure 4: The average reward and win rate of the aligned models improve as we increase the sample size $M$ used for approximating the vBo$N$ loss function.

between different prompts to learn an approximation to $\log \mathrm{F}$ function, which we plan as future work.

## 6 Summarization

We further employ variational Bo$N$ in a summarization task, where the goal is to generate summaries that align with human preferences. The reference model, $\pi_{\mathrm{ref}}$, is a `pythia-2.8B` model fine-tuned on human-written summaries of Reddit posts Stiennon et al. (2020).[15] We use SFT to refer to this model in the plots. We use two separate reward models for training and evaluation: a `pythia-2.8B`[16] reward model for fine-tuning and a larger `pythia-6.9B`[17] model exclusively for evaluation.

**Dataset.** To evaluate the generalization ability of the aligned models on out-of-distribution data, we fine-tune the models using only posts from the `relationship` and `relationship_advice` subreddits

---

[14]This is particularly advantageous since practitioners often perform the optimization multiple times to test various hyperparameter settings.

[15]We use `https://huggingface.co/cleanrl/EleutherAI_pythia-2.8b-deduped__sft__tldr`.

[16]We use `https://huggingface.co/cleanrl/EleutherAI_pythia-2.8b-deduped__reward__tldr`.

[17]We use `https://huggingface.co/cleanrl/EleutherAI_pythia-6.9b-deduped__reward__tldr`.

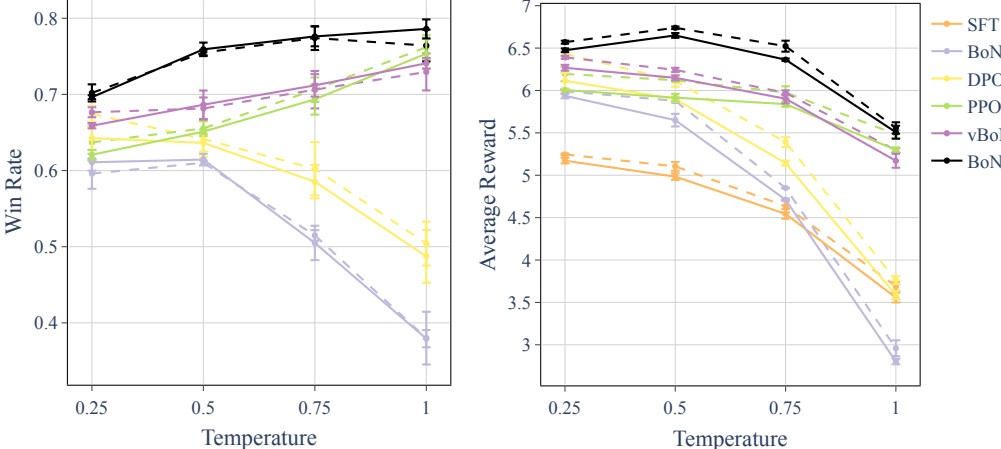

(a) Comparing the win rates of alignment methods against samples from the $\pi_{\text{ref}}$. vBoN achieves closer results to BoN compared to other alignment-via-fine-tuning methods.

(b) Comparing the average rewards obtained from the evaluator reward model. BoN outperforms other alignment methods, and vBoN achieves closer results to BoN compared to other alignment-via-fine-tuning methods.

Figure 5: Performance of different alignment methods on the summarization task. Solid traces show the performance on in-distribution Reddit posts, while dashed lines demonstrate the out-of-distribution performance. Overall, BoN is the most effective approach in achieving high win rates and average rewards across all sampling temperatures. Our variational approximation to BoN (vBoN) gets closest to the performance of BoN while being significantly cheaper at inference time.

of the `Reddit TL;DR` (Stiennon et al., 2020) dataset. We then assess the models' performance on the two types of data by dividing the test set into two equally-sized groups: in-distribution Reddit posts from the `relationship` and `relationship_advice` subreddits, and out-of-distribution posts from the rest of the subreddits. We visualize the performance of methods on in-distribution data with a solid trace and on out-of-distribution data with a dashed trace.

**Experimental setup.** We fine-tune the model with both the KL-constrained RL objective and vBoN objective for 10000 episodes. Similar to the previous experiment, we use 200 samples to estimate $\log \text{F}(\cdot)$ values. To create a smooth and continuous reward function, we further fit an exponential curve[18] to the estimates. We set $N = 100$ for BoN and vBoN methods and the equivalent value of $\beta = 0.05$ for the KL-constrained RL objective. We closely follow Huang et al. (2024) for setting the hyperparameters of the PPO algorithm; please refer to App. F for more experimental details. After fine-tuning, we sample from the aligned models with different sampling temperatures $t \in [0.25, 0.5, 0.75, 1.]$, each with 3 different random seeds.

**Win rates.** In Fig. 5a, we visualize the average and standard deviation of win rates compared against the samples from the SFT model. Notably, BoN achieves the highest win rates, which is consistent with findings from previous studies (Rafailov et al., 2023). We do not observe any significant differences between BoN performance on in-distribution (solid trace) and out-of-distribution data,[19] which is expected as BoN is an alignment-via-inference method. Similarly, we mostly do not observe significant differences between in- and out-of-distribution performance of all alignment-via-fine-tuning methods, indicating that these methods can generalize effectively in this experimental setup. DPO and BoNBoN only manage to perform competitively to other methods at lower temperatures (0.25, 0.5), and their performance drops significantly at higher temperatures (0.75, 1). Importantly, while PPO and vBoN perform comparably at higher temperatures, vBoN significantly outperforms PPO at lower temperatures (0.25 and 0.5).

---

[18]We fit an exponential function of the form $f(x) = -a \exp(-bx)$ to the data using non-linear least squares.

[19]The difference between the two data distributions becomes more apparent at temperature 1, potentially due to increased sample diversity in this setting.

Table 2: An example of summaries sampled at temperature $0.5$ and their corresponding reward obtained from the evaluator reward model.

| Content | Reward |
|---|---|
| SUBREDDIT: r/relationship_advice
TITLE: Stuck in a rut and in need of advice/inspiration!
POST: My boyfriend and I have been together for 3 years, and living together for 2. I'm quite the homebody, and when we first met, he was very outgoing and loved partying and socialising (although he was a student at the time). We're both working now, and most nights we find ourselves doing the same things: watching series (luckily we enjoy the same shows), playing Minecraft or playing various board games. We're tired after work, and can't bring ourselves to leave the house. The weekend is much the same – lots of sleep, or sitting around staring at one screen or another. We do party occasionally (we'll head to a pub once every few months) and there are a few mutual friends we enjoy spending time with, but I worry that we've become stuck in our boring ways. I really enjoy our lifestyle, and would be quite happy to never leave the house again, but I'm starting to feel guilty for turning him into a 50 year-old when he's only 24. Any ideas for shaking things up a little? Bear in mind that we live in a small town in South Africa, and neither of us has a car. | - |
| SFT: I'm stuck in a rut, and need to shake things up to see if it'll work out. Any advice? | 3.08 |
| PPO: In need of inspiration to break out of rut and live life fully! Any ideas welcome! | 4.59 |
| vBo$N$: Been happily living together for 2yr+, feeling bored after work regularly, looking for ideas to spice things up! | 6.79 |
| Bo$N$: My boyfriend and I have been together for 3 years, and are both working full time. We spend most of our time in the house, and have become boring. What can we do to shake things up? | 9.18 |

**Average rewards.** In Fig. 5b, we measure the average rewards across different temperatures. As the temperature increases, the average reward decreases consistently across all methods. This trend is also evident in the qualitative analysis in App. I, where we show sampled summaries at different temperatures. DPO and BoNBoN suffer more from increasing the temperature, as the average rewards get close to (or even worse than) the SFT average rewards. Generally, the average reward results align with the win-rate trends, and we observe that vBo$N$ achieves significantly higher rewards compared to PPO at lower temperatures. In Tab. 2, we show an example of summaries generated from the fine-tuned models with their associated reward values.

## 7 CONCLUSION

Motivated by the effectiveness of the Bo$N$ algorithm, we formally derive a variational approximation to the distribution induced by Bo$N$ algorithm via fine-tuning language models. Our analysis highlights the similarities and distinctions between the variational Bo$N$ objective and the KL-constrained RL objectives. Our empirical findings reveal that models fine-tuned using the variational approximation to Bo$N$ not only attain high reward values but also maintain proximity to the reference models. Crucially, inference on the fine-tuned models with the vBo$N$ objective remains as cost-effective as inference on the original reference model.

## ACKNOWLEDGEMENTS

We thank Ahmad Beirami for the fruitful discussion in the early stages of this project. We also thank Amrit Singh Bedi for identifying a typo in a previous version of the bound derivations. Finally, we thank the anonymous reviewers for their feedback. Afra Amini is supported by the ETH AI Center doctoral fellowship.

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

| Symbol | Type | Explanation |
|:---:|:---:|:---|
| $\Sigma$ | alphabet | $\Sigma$ is a set of symbols |
| $\boldsymbol{y}, \boldsymbol{y}'$ | $\in \Sigma^*$ | strings in $\Sigma^*$ |
| $\boldsymbol{x}$ | $\in \Sigma^*$ | prompt string in $\Sigma^*$ |
| $\boldsymbol{\theta}$ | $\in \boldsymbol{\Theta}$ | A real vector representing the parameters of a language model |
| $\pi_{\boldsymbol{\theta}}$ | language model | A language model parameterized by $\boldsymbol{\theta}$ |
| $\pi_{\text{ref}}$ | language model | A supervised-fine-tuned language model |
| $r$ | $\Sigma^* \to \mathbb{R}$ | A reward model |
| $\beta$ | $\mathbb{R}$ | Regularization parameter for the KL divergence term |
| F | $\mathbb{R} \to \mathbb{R}$ | A strict cumulative density function of reward values under $\pi_{\text{ref}}$ |
| $N$ | $\mathbb{Z}^+$ | Number of samples used in BoN algorithm |
| $M$ | $\mathbb{Z}^+$ | Number of samples used in the MC estimator |

Table 3: A summary of the notation used in the paper

# A RELATED WORK

**Best-of-$N$.** BoN is a straightforward alignment-via-inference algorithm to optimize the output of the language model using a trained reward model (Charniak & Johnson, 2005; Stiennon et al., 2020). Despite its simplicity, BoN performs comparably or even better than other alignment methods, such as RLHF and direct preference optimization (Nakano et al., 2022; Gao et al., 2023; Rafailov et al., 2023). However, as noted by Stiennon et al. (2020), BoN is an inefficient algorithm due to the reduced throughput at inference time.

**Applications.** BoN has been applied successfully at various stages of the development of language models. Meta (2023); Dong et al. (2023) employ iterative supervised fine-tuning on the outputs of the BoN algorithm to clone its behavior in the model. Pace et al. (2024) leverage BoN to enhance reward modeling by training the reward model on both the best and worst responses. Additionally, Brown et al. (2024); Snell et al. (2024) explore the scaling laws for alignment-via-inference methods and demonstrate how to utilize the limited inference budget to achieve the alignment.

**Best-of-$N$ as an alignment-via-fine-tuning method.** Two concurrent efforts to ours have also attempted to convert BoN to an alignment-via-fine-tuning method. First, Gui et al. (2024) approximate the BoN by maximizing the likelihood of the Best-of-$N$ response and adjusting the relative likelihood of the Best-of-$N$ and the Worst-of-$N$ response. Second, Sessa et al. (2024), similar to ours, uses reinforcement learning to minimize the distance between the language model and the BoN policy. Different from ours, and to reduce the fine-tuning time, the authors use a crude estimation of $\log F$ and approximate the distance to Best-of-$N$ by iteratively distilling the Best-of-2 model as a moving anchor.

# B PROOF OF PROP. 1

**Proposition 1.** *Suppose $r: \Sigma^* \to \mathbb{R}$ is a one-to-one mapping. Then, the probability of a string $\boldsymbol{y}$ under $\pi_{bon}$ is given by*

$$\pi_{bon}(\boldsymbol{y}) = \sum_{i=1}^{N} \binom{N}{i} F\big(r(\boldsymbol{y})\big)^{N-i} \pi_{\text{ref}}(\boldsymbol{y})^i, \qquad F\big(r(\boldsymbol{y})\big) \stackrel{\text{def}}{=} \mathbb{P}_{\boldsymbol{y}' \sim \pi_{\text{ref}}} \big(r(\boldsymbol{y}') < r(\boldsymbol{y})\big). \qquad (4)$$

*Proof.* The proof follows Casella & Berger (2001, Theorem 5.4.3). To compute $\pi_{\text{bon}}(\boldsymbol{y})$, we first define two events: (i) the event that all $N$ samples have rewards less than or equal to $r(\boldsymbol{y})$, and (ii) the

event that all $N$ samples have rewards less than $r(\boldsymbol{y})$. The probability of those events is as follows:[20]

$$p_1(\boldsymbol{y}) \stackrel{\text{def}}{=} \mathbb{P}(\text{all } N \text{ samples have rewards} \leq r(\boldsymbol{y})) = \Big( \mathrm{F}\big(r(\boldsymbol{y})\big) + \pi_{\text{ref}}(\boldsymbol{y}) \Big)^N \tag{10a}$$

$$p_2(\boldsymbol{y}) \stackrel{\text{def}}{=} \mathbb{P}(\text{all } N \text{ samples have rewards} < r(\boldsymbol{y})) = \mathrm{F}\big(r(\boldsymbol{y})\big)^N. \tag{10b}$$

Note that for Eq. (10a) to hold, we need the assumption that the reward function is a one-to-one mapping.[21] Furthermore, given this assumption, $\pi_{\text{bon}}(\boldsymbol{y})$ is the probability that *at least* one of the sampled strings out of $N$ samples have the reward exactly equal to $r(\boldsymbol{y})$ and the rest of the samples have rewards less than or equal to $r(\boldsymbol{y})$. Given how we defined $p_1$ and $p_2$, we have $\pi_{\text{bon}}(\boldsymbol{y}) = p_1(\boldsymbol{y}) - p_2(\boldsymbol{y})$.

$$\pi_{\text{bon}}(\boldsymbol{y}) = \Big( \mathrm{F}\big(r(\boldsymbol{y})\big) + \pi_{\text{ref}}(\boldsymbol{y}) \Big)^N - \mathrm{F}\big(r(\boldsymbol{y})\big)^N = \sum_{i=1}^{N} \binom{N}{i} \mathrm{F}\big(r(\boldsymbol{y})\big)^{N-i} \pi_{\text{ref}}(\boldsymbol{y})^i. \tag{11}$$

∎

## C    Strategies for Non-Injective Reward Functions

If the reward function is not injective, we need a tie-breaking strategy for the BoN algorithm. We formalize this as defining a total order $\prec_r$ on $\Sigma^*$ as follows: for any two strings $\boldsymbol{y}_1$ and $\boldsymbol{y}_2$, if $r(\boldsymbol{y}_1) < r(\boldsymbol{y}_2)$ then we have $\boldsymbol{y}_1 \prec_r \boldsymbol{y}_2$. If $r(\boldsymbol{y}_1) = r(\boldsymbol{y}_2)$ then $\boldsymbol{y}_1 \prec_r \boldsymbol{y}_2$ only if $\boldsymbol{y}_1 \prec \boldsymbol{y}_2$, where $\prec$ is some arbitrary but fixed total order, e.g., lexicographic order. Therefore, we define $\mathrm{F}(\boldsymbol{y})$ as

$$\mathrm{F}(\boldsymbol{y}) \stackrel{\text{def}}{=} \mathbb{P}\big(\boldsymbol{y}' \prec_r \boldsymbol{y}\big). \tag{12}$$

We then need to define the two events and their probabilities, $p_1$ and $p_2$, given this total order on strings, as follows:

$$p_1(\boldsymbol{y}) \stackrel{\text{def}}{=} \mathbb{P}(\text{all } N \text{ samples are } \preceq_r \boldsymbol{y}) = \Big( \mathrm{F}(\boldsymbol{y}) + \pi_{\text{ref}}(\boldsymbol{y}) \Big)^N \tag{13a}$$

$$p_2(\boldsymbol{y}) \stackrel{\text{def}}{=} \mathbb{P}(\text{all } N \text{ samples are } \prec_r \boldsymbol{y}) = \mathrm{F}(\boldsymbol{y})^N \tag{13b}$$

The rest of the proof is the same as with the one-to-one reward functions.

## D    Proof of Thm. 2

**Theorem 2.** *We have $\mathcal{J}^{\text{vBoN}}(\boldsymbol{\theta}) \geq L(\boldsymbol{\theta})$, where*

$$L(\boldsymbol{\theta}) \stackrel{\text{def}}{=} (N-1) \mathop{\mathbb{E}}_{\boldsymbol{y} \sim \pi_{\boldsymbol{\theta}}} \Big[ \log \mathrm{F}\big(r(\boldsymbol{y})\big) \Big] - D_{\text{KL}}\big(\pi_{\boldsymbol{\theta}} \,\|\, \pi_{\text{ref}}\big). \tag{8}$$

---

[20]The PMF of BoN is also derived by Beirami et al. (Lemma 1; 2024). In their notation, $p_1 = \mathcal{F}$ and $p_2 = \mathcal{F}^{-1}$.
[21]If the reward function is not a one-to-one mapping, we need to devise a tie-breaking strategy. See App. C for further discussion.

*Proof.* First, we prove $\mathcal{J}^{\text{vBoN}}(\boldsymbol{\theta}) \geq L(\boldsymbol{\theta})$.

$$D_{\text{KL}}\big(\pi_{\boldsymbol{\theta}} \,\|\, \pi_{\text{bon}}\big) = \mathop{\mathbb{E}}_{\boldsymbol{y} \sim \pi_{\boldsymbol{\theta}}} \Big[ \log \pi_{\boldsymbol{\theta}}(\boldsymbol{y}) - \log \pi_{\text{bon}}(\boldsymbol{y}) \Big] \tag{14a}$$

$$= \mathop{\mathbb{E}}_{\boldsymbol{y} \sim \pi_{\boldsymbol{\theta}}} \Big[ \log \pi_{\boldsymbol{\theta}}(\boldsymbol{y}) - \log \sum_{i=1}^{N} \binom{N}{i} \text{F}\big(r(\boldsymbol{y})\big)^{N-i} \pi_{\text{ref}}(\boldsymbol{y})^{i} \Big] \tag{14b}$$

$$\leq \mathop{\mathbb{E}}_{\boldsymbol{y} \sim \pi_{\boldsymbol{\theta}}} \Big[ \log \pi_{\boldsymbol{\theta}}(\boldsymbol{y}) - \log \sum_{i=1}^{N=1} \binom{N}{i} \text{F}\big(r(\boldsymbol{y})\big)^{N-i} \pi_{\text{ref}}(\boldsymbol{y})^{i} \Big] \tag{14c}$$

$$\leq \mathop{\mathbb{E}}_{\boldsymbol{y} \sim \pi_{\boldsymbol{\theta}}} \Big[ \log \pi_{\boldsymbol{\theta}}(\boldsymbol{y}) - \log N \, \text{F}\big(r(\boldsymbol{y})\big)^{N-1} \pi_{\text{ref}}(\boldsymbol{y})^{1} \Big] \tag{14d}$$

$$\leq \mathop{\mathbb{E}}_{\boldsymbol{y} \sim \pi_{\boldsymbol{\theta}}} \Big[ \log \pi_{\boldsymbol{\theta}}(\boldsymbol{y}) - \log \text{F}\big(r(\boldsymbol{y})\big)^{N-1} \pi_{\text{ref}}(\boldsymbol{y}) \Big] \tag{14e}$$

$$= \mathop{\mathbb{E}}_{\boldsymbol{y} \sim \pi_{\boldsymbol{\theta}}} \Big[ \log \pi_{\boldsymbol{\theta}}(\boldsymbol{y}) - \log \pi_{\text{ref}}(\boldsymbol{y}) - (N-1) \log \text{F}\big(r(\boldsymbol{y})\big) \Big] \tag{14f}$$

$$= D_{\text{KL}}\big(\pi_{\boldsymbol{\theta}} \,\|\, \pi_{\text{ref}}\big) - (N-1) \mathop{\mathbb{E}}_{\boldsymbol{y} \sim \pi_{\boldsymbol{\theta}}} \Big[ \log \text{F}\big(r(\boldsymbol{y})\big) \Big] \stackrel{\text{def}}{=} -L(\boldsymbol{\theta}). \tag{14g}$$

The inequality in Eq. (14c) stems from the fact that we drop positive terms in the summation and only keep the first term. Therefore, the lower bound for our objective is:

$$\mathcal{J}^{\text{vBoN}}(\boldsymbol{\theta}) = -D_{\text{KL}}\big(\pi_{\boldsymbol{\theta}} \,\|\, \pi_{\text{bon}}\big) \geq (N-1) \mathop{\mathbb{E}}_{\boldsymbol{y} \sim \pi_{\boldsymbol{\theta}}} \Big[ \log \text{F}\big(r(\boldsymbol{y})\big) \Big] - D_{\text{KL}}\big(\pi_{\boldsymbol{\theta}} \,\|\, \pi_{\text{ref}}\big). \tag{15}$$

∎

Another approach to deriving a lower bound is by using Jensen's inequality. By doing so, we arrive at the following theorem.

**Theorem 3.** *Let $\alpha = \frac{(N+2)(N-1)}{2}$, $\beta = \frac{N(N+1)}{2}$, and $\gamma = \frac{N(N-1)}{2}$. Then, we have $\mathcal{J}^{\text{vBoN}}(\boldsymbol{\theta}) \geq L_1(\boldsymbol{\theta})$, where we further define*

$$L_1(\boldsymbol{\theta}) \stackrel{\text{def}}{=} \gamma \mathop{\mathbb{E}}_{\boldsymbol{y} \sim \pi_{\boldsymbol{\theta}}} \Big[ \log \text{F}\big(r(\boldsymbol{y})\big) \Big] - \alpha \text{H}\big(\pi_{\boldsymbol{\theta}}\big) - \beta D_{\text{KL}}\big(\pi_{\boldsymbol{\theta}} \,\|\, \pi_{\text{ref}}\big). \tag{16}$$

*Proof.*

$$D_{\mathrm{KL}}\big(\pi_{\boldsymbol{\theta}} \mid\mid \pi_{\mathrm{bon}}\big) = \underset{\boldsymbol{y} \sim \pi_{\boldsymbol{\theta}}}{\mathbb{E}}\Big[\log \pi_{\boldsymbol{\theta}}(\boldsymbol{y}) - \log \pi_{\mathrm{bon}}(\boldsymbol{y})\Big] \tag{17a}$$

$$= \underset{\boldsymbol{y} \sim \pi_{\boldsymbol{\theta}}}{\mathbb{E}}\Big[\log \pi_{\boldsymbol{\theta}}(\boldsymbol{y}) - \log \sum_{i=1}^{N}\binom{N}{i}\mathrm{F}\big(r(\boldsymbol{y})\big)^{N-i}\pi_{\mathrm{ref}}(\boldsymbol{y})^{i}\Big] \tag{17b}$$

$$\leq \underset{\boldsymbol{y} \sim \pi_{\boldsymbol{\theta}}}{\mathbb{E}}\Big[\log \pi_{\boldsymbol{\theta}}(\boldsymbol{y}) - \sum_{i=1}^{N}\log\binom{N}{i}\mathrm{F}\big(r(\boldsymbol{y})\big)^{N-i}\pi_{\mathrm{ref}}(\boldsymbol{y})^{i}\Big] \tag{17c}$$

$$= \underset{\boldsymbol{y} \sim \pi_{\boldsymbol{\theta}}}{\mathbb{E}}\Big[\log \pi_{\boldsymbol{\theta}}(\boldsymbol{y}) - \sum_{i=1}^{N}\log\binom{N}{i} - \sum_{i=1}^{N}\log \mathrm{F}\big(r(\boldsymbol{y})\big)^{N-i} - \sum_{i=1}^{N}\log \pi_{\mathrm{ref}}(\boldsymbol{y})^{i}\Big] \tag{17d}$$

$$= \underset{\boldsymbol{y} \sim \pi_{\boldsymbol{\theta}}}{\mathbb{E}}\Big[\log \pi_{\boldsymbol{\theta}}(\boldsymbol{y}) - \sum_{i=1}^{N}\log\binom{N}{i} - \log \mathrm{F}\big(r(\boldsymbol{y})\big)\sum_{i=1}^{N}(N-i) - \log \pi_{\mathrm{ref}}(\boldsymbol{y})\sum_{i=1}^{N}i\Big] \tag{17e}$$

$$\leq \underset{\boldsymbol{y} \sim \pi_{\boldsymbol{\theta}}}{\mathbb{E}}\Big[\log \pi_{\boldsymbol{\theta}}(\boldsymbol{y}) - \frac{N(N-1)}{2}\log \mathrm{F}\big(r(\boldsymbol{y})\big) - \frac{N(N+1)}{2}\log \pi_{\mathrm{ref}}(\boldsymbol{y})\Big] \tag{17f}$$

$$= \underset{\boldsymbol{y} \sim \pi_{\boldsymbol{\theta}}}{\mathbb{E}}\Big[\log \pi_{\boldsymbol{\theta}}(\boldsymbol{y}) - \frac{N(N+1)}{2}\log \pi_{\mathrm{ref}}(\boldsymbol{y}) - \frac{N(N-1)}{2}\log \mathrm{F}\big(r(\boldsymbol{y})\big)\Big] \tag{17g}$$

$$= \frac{N(N+1)}{2}D_{\mathrm{KL}}\big(\pi_{\boldsymbol{\theta}} \mid\mid \pi_{\mathrm{ref}}\big) + \underset{\pi_{\boldsymbol{\theta}}}{\mathbb{E}}\Big[\frac{-(N+2)(N-1)}{2}\log \pi_{\boldsymbol{\theta}}(\boldsymbol{y}) - \frac{N(N-1)}{2}\log \mathrm{F}\big(r(\boldsymbol{y})\big)\Big] \tag{17h}$$

$$= \frac{N(N+1)}{2}D_{\mathrm{KL}}\big(\pi_{\boldsymbol{\theta}} \mid\mid \pi_{\mathrm{ref}}\big) + \frac{(N+2)(N-1)}{2}\mathrm{H}\big(\pi_{\boldsymbol{\theta}}\big) - \underset{\pi_{\boldsymbol{\theta}}}{\mathbb{E}}\Big[\frac{N(N-1)}{2}\log \mathrm{F}\big(r(\boldsymbol{y})\big)\Big] \tag{17i}$$

In Eq. (17c), because $-\log(x)$ is convex for $x \geq 0$, we applied Jensen's inequality to obtain the upper bound. Abstracting away from the three multiplicative factors, naming them $\gamma$, $\alpha$ and $\beta$, we end up with the following function

$$\mathcal{J}^{\mathrm{vBON}}(\boldsymbol{\theta}) = -D_{\mathrm{KL}}\big(\pi_{\boldsymbol{\theta}} \mid\mid \pi_{\mathrm{bon}}\big) \geq \gamma \underset{\boldsymbol{y} \sim \pi_{\boldsymbol{\theta}}}{\mathbb{E}}\log \mathrm{F}\big(r(\boldsymbol{y})\big) - \alpha \mathrm{H}(\pi_{\boldsymbol{\theta}}) - \beta D_{\mathrm{KL}}\big(\pi_{\boldsymbol{\theta}} \mid\mid \pi_{\mathrm{ref}}\big), \tag{18}$$

which is a bound for some settings of $\gamma$, $\alpha$ and $\beta$. ■

Importantly, $L_1$ is a looser bound compared to $L$. We formalize this in the following theorem.

**Theorem 4.** *For every $\boldsymbol{\theta} \in \boldsymbol{\Theta}$, we have $L(\boldsymbol{\theta}) \geq L_1(\boldsymbol{\theta})$.*

*Proof.* We prove $-L_1(\boldsymbol{\theta}) \geq -L(\boldsymbol{\theta})$, meaning that $L$ is a tighter lower bound. According to Eq. (17f), we have:

$$-L_1(\boldsymbol{\theta}) \geq \underset{\boldsymbol{y} \sim \pi_{\boldsymbol{\theta}}}{\mathbb{E}}\Big[\log \pi_{\boldsymbol{\theta}}(\boldsymbol{y}) - \sum_{i=1}^{N}\log \mathrm{F}\big(r(\boldsymbol{y})\big)^{N-i}\pi_{\mathrm{ref}}(\boldsymbol{y})^{i}\Big] \tag{19a}$$

$$\geq \underset{\boldsymbol{y} \sim \pi_{\boldsymbol{\theta}}}{\mathbb{E}}\Big[\log \pi_{\boldsymbol{\theta}}(\boldsymbol{y}) - \sum_{i=1}^{N=1}\log \mathrm{F}\big(r(\boldsymbol{y})\big)^{N-i}\pi_{\mathrm{ref}}(\boldsymbol{y})^{i}\Big] \tag{19b}$$

$$= \underset{\boldsymbol{y} \sim \pi_{\boldsymbol{\theta}}}{\mathbb{E}}\Big[\log \pi_{\boldsymbol{\theta}}(\boldsymbol{y}) - \log \mathrm{F}\big(r(\boldsymbol{y})\big)^{N-1}\pi_{\mathrm{ref}}(\boldsymbol{y})\Big] = -L(\boldsymbol{\theta}). \tag{19c}$$

■

| Hypterparameter | Value |
|---|---|
| Episodes | 10000 |
| Optimizer | AdamW ($\epsilon = 1e-5$, lr$= 3e-6$) |
| Scheduler | Linear |
| Batch Size | 32 |
| $\beta$ (Both for vBo$N$ and KL-constrained RL objective) | 0.05 |
| $\gamma$ (Discount Factor) | 1 |
| $\lambda$ (for GAE) | 0.95 |
| Number of PPO Update Iteration Per Epoch | 4 |
| PPO's Policy Clipping Coefficient | 0.2 |
| Value Clipping Coefficient | 0.2 |
| Value Function Coefficient | 0.2 |
| Value Function Loss Clipping | True |
| Sampling Temperature | 0.7 |

## E   vBo$N$ Pseudocode

---
**Algorithm 1** The vBo$N$ algorithm

---
1: **procedure** VBO$N(\pi_{\text{ref}}, r, N, E, B)$     ▷ $\mathcal{D}$: *the prompt dataset, E: number of epochs, B batch size*
2:     Initialize $\pi_{\boldsymbol{\theta}}$ with $\pi_{\text{ref}}$
3:     **for** $E$ epochs :
4:       **for** each batch in $\mathcal{D}$ :
5:         $\boldsymbol{y}^{(1)}, ... , \boldsymbol{y}^{(B)} \sim \pi_{\boldsymbol{\theta}}(\cdot)$     ▷ *Sample 1 response for each prompt in the batch*
6:         Compute $r(\boldsymbol{y}^{(1)}), ... , r(\boldsymbol{y}^{(B)})$
7:         Compute $\text{F}\big(r(\boldsymbol{y}^{(1)})\big), ... , \text{F}\big(r(\boldsymbol{y}^{(B)})\big)$
8:         Optimize $\pi_{\boldsymbol{\theta}}$ with Eq. (5) (or Eq. (8)) using PPO
9:     **return** $\pi_{\boldsymbol{\theta}}$

---

## F   Experimental Details

**Hyperparameter sweep in the sentiment experiment.**   To visualize the trade-off between the expected rewards and KL divergence, we vary the degree of the visualization using the following hyperparameters for each method:

- **Bo$N$-SFT**: $N \in [10, 50, 90, 130, 170, 210, 250, 290, 330, 370, 410, 450, 490, 530, 570, 600]$ with 2 different seeds, resulting in 32 runs.
- **PPO**: $\beta \in [0.005, 0.01, 0.02, 0.03, 0.04, 0.05, 0.1, 0.2, 0.3, 0.4, 0.5, 1., 2., 3., 4., 5.]$ with 2 different seeds, resulting in 32 runs.
- **DPO**: $\beta \in [0.01, 0.1, 0.2, 0.3, 0.4, 0.5, 1., 2., 3., 4., 5.]$ with 3 different seeds, resulting in 33 runs.
- **BoNBoN** and **vBo$N$**: $N \in [1, 2, 3, 4, 8, 16, 32, 64, 128, 256, 512]$ with 3 different seeds, resulting in 33 runs.
- **vBo$N$** with $L$ bound: $\beta \in [0.005, 0.01, 0.02, 0.03, 0.04, 0.05, 0.1, 0.2, 0.3, 0.4, 0.5, 1., 2., 3., 4., 5.]$ with 2 different seeds, resulting in 32 runs. Note that comparing Eq. (5) and Eq. (1), we have $N = \frac{1}{\beta} + 1$.

**PPO hyperparameters.**   In App. F, we include the hyperparameters used with the PPO algorithm for the summarization experiment.

## G   Comparing the vBo$N$ Objective and $L$ Lower Bound

We compare the performance of models fine-tuned with the vBo$N$ objective and its lower bound ($L$) in Fig. 6. We observe that the performance of the models is very close to each other.

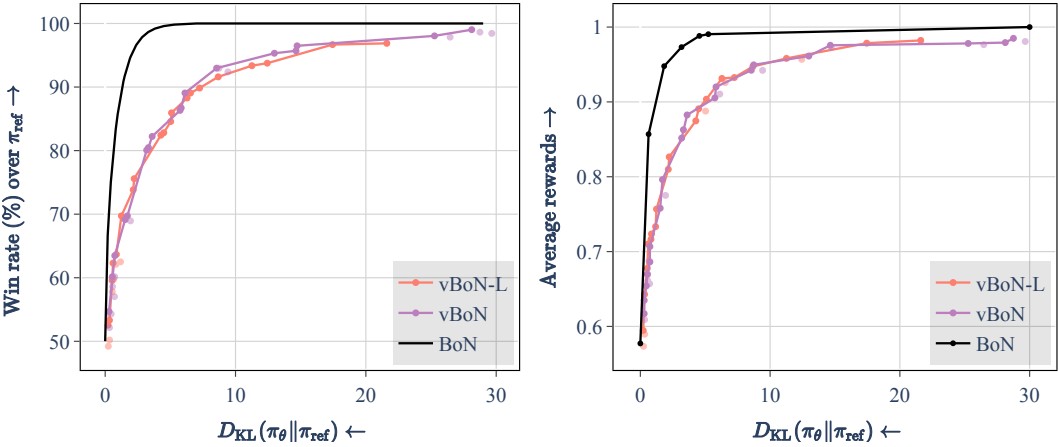

Figure 6: Comparing models trained with the vBo$N$ objective and its lower bound ($L$). We observe that the performance of the two methods is very close to each other.

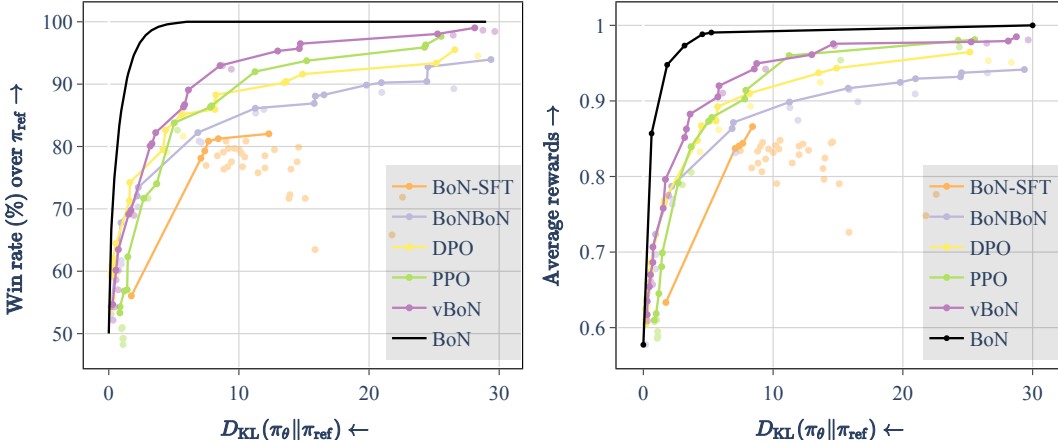

(a) 4% of points on Pareto front belong to BoNBoN, 4% to PPO, 42% to DPO, and 50% to vBo$N$.

(b) 7% of points on Pareto from belong to BoNBoN, 10% DPO, 33% PPO, and 50% vBo$N$.

Figure 7: Steering generated movie reviews towards positive sentiment. Points that are not on the Pareto front have lower opacity.

## H  ADDITIONAL EXPERIMENTS WITH BO$N$-SFT

We further experiment with training with the maximum likelihood objective on Bo$N$ generations when varying $N$. The results are depicted in Fig. 7. We observe that Bo$N$ diverges too much from the reference model compared to other fine-tuning methods for alignment.

## I  QUALITATIVE RESULTS

Table 4: An example of summaries sampled at temperature 0.25 and their corresponding reward obtained from the evaluator reward model.

| Content | Reward |
|---|---|
| SUBREDDIT: r/relationship_advice
TITLE: Stuck in a rut and in need of advice/inspiration!
POST: My boyfriend and I have been together for 3 years, and living together for 2. I'm quite the homebody, and when we first met, he was very outgoing and loved partying and socialising (although he was a student at the time). We're both working now, and most nights we find ourselves doing the same things: watching series (luckily we enjoy the same shows), playing Minecraft or playing various board games. We're tired after work, and can't bring ourselves to leave the house. The weekend is much the same – lots of sleep, or sitting around staring at one screen or another. We do party occasionally (we'll head to a pub once every few months) and there are a few mutual friends we enjoy spending time with, but I worry that we've become stuck in our boring ways. I really enjoy our lifestyle, and would be quite happy to never leave the house again, but I'm starting to feel guilty for turning him into a 50 year-old when he's only 24. Any ideas for shaking things up a little? Bear in mind that we live in a small town in South Africa, and neither of us has a car. | - |
| SFT: I'm stuck in a rut with my boyfriend, and I'd like to shake things up a little. Any ideas? | 5.83 |
| PPO: In need of inspiration for moving past boring routines and becoming active and open minded individuals. Any ideas welcome! | 4.88 |
| vBo$N$: Been dating for three years and living together for two. Feeling stuck in rut after working hard all week. Looking for ideas to shake things up! | 7.39 |
| Bo$N$: Been together for 3 years, and living together for 2. We're both working now, and find ourselves doing the same things: watching series, playing games, or having sex. We're tired after work, and can't bring ourselves to leave | 7.14 |

Table 5: An example of summaries sampled at temperature 0.5 and their corresponding reward obtained from the evaluator reward model.

| Content | Reward |
|---|---|
| SUBREDDIT: r/relationship_advice
TITLE: Stuck in a rut and in need of advice/inspiration!
POST: My boyfriend and I have been together for 3 years, and living together for 2. I'm quite the homebody, and when we first met, he was very outgoing and loved partying and socialising (although he was a student at the time). We're both working now, and most nights we find ourselves doing the same things: watching series (luckily we enjoy the same shows), playing Minecraft or playing various board games. We're tired after work, and can't bring ourselves to leave the house. The weekend is much the same – lots of sleep, or sitting around staring at one screen or another. We do party occasionally (we'll head to a pub once every few months) and there are a few mutual friends we enjoy spending time with, but I worry that we've become stuck in our boring ways. I really enjoy our lifestyle, and would be quite happy to never leave the house again, but I'm starting to feel guilty for turning him into a 50 year-old when he's only 24. Any ideas for shaking things up a little? Bear in mind that we live in a small town in South Africa, and neither of us has a car. | - |
| SFT: I'm stuck in a rut, and need to shake things up to see if it'll work out. Any advice? | 3.08 |
| PPO: In need of inspiration to break out of rut and live life fully! Any ideas welcome! | 4.59 |
| vBo$N$: Been happily living together for 2yr+, feeling bored after work regularly, looking for ideas to spice things up! | 6.79 |
| Bo$N$: My boyfriend and I have been together for 3 years, and are both working full time. We spend most of our time in the house, and have become boring. What can we do to shake things up? | 9.18 |

Table 6: An example of summaries sampled at temperature $0.75$ and their corresponding reward obtained from the evaluator reward model.

| Content | Reward |
|---|---|
| SUBREDDIT: r/relationship_advice
TITLE: Stuck in a rut and in need of advice/inspiration!
POST: My boyfriend and I have been together for 3 years, and living together for 2. I'm quite the homebody, and when we first met, he was very outgoing and loved partying and socialising (although he was a student at the time). We're both working now, and most nights we find ourselves doing the same things: watching series (luckily we enjoy the same shows), playing Minecraft or playing various board games. We're tired after work, and can't bring ourselves to leave the house. The weekend is much the same – lots of sleep, or sitting around staring at one screen or another. We do party occasionally (we'll head to a pub once every few months) and there are a few mutual friends we enjoy spending time with, but I worry that we've become stuck in our boring ways. I really enjoy our lifestyle, and would be quite happy to never leave the house again, but I'm starting to feel guilty for turning him into a 50 year-old when he's only 24. Any ideas for shaking things up a little? Bear in mind that we live in a small town in South Africa, and neither of us has a car. | - |
| SFT: Boyfriend and I are stuck in a rut. We don't do anything fun, and I'm starting to feel guilty for not even trying to get away from it all. | 4.41 |
| PPO: In rutting boredom with boring life; any ideas for shaking things up a little? Any ideas for improving relationships? | 4.76 |
| vBo$N$: Been happily cohabiting for three years with boyfriend of two years. Feeling guilty for turning him into an old man! Any ideas for shaking things up? | 6.02 |
| Bo$N$: Boyfriend and I are a little boring, and want to shake things up so we can go out and have fun but we're too stuck on the idea of staying in all day. | 5.47 |

Table 7: An example of summaries sampled at temperature $1.$ and their corresponding reward obtained from the evaluator reward model.

| Content | Reward |
|---|---|
| SUBREDDIT: r/relationship_advice
TITLE: Stuck in a rut and in need of advice/inspiration!
POST: My boyfriend and I have been together for 3 years, and living together for 2. I'm quite the homebody, and when we first met, he was very outgoing and loved partying and socialising (although he was a student at the time). We're both working now, and most nights we find ourselves doing the same things: watching series (luckily we enjoy the same shows), playing Minecraft or playing various board games. We're tired after work, and can't bring ourselves to leave the house. The weekend is much the same – lots of sleep, or sitting around staring at one screen or another. We do party occasionally (we'll head to a pub once every few months) and there are a few mutual friends we enjoy spending time with, but I worry that we've become stuck in our boring ways. I really enjoy our lifestyle, and would be quite happy to never leave the house again, but I'm starting to feel guilty for turning him into a 50 year-old when he's only 24. Any ideas for shaking things up a little? Bear in mind that we live in a small town in South Africa, and neither of us has a car. | - |
| SFT: My boyfriend and I used to party all the time and are ending up reverting to the boring old lifestyle of a 24-year-old. How can we both break out of a rut and get some fun back in our relationship? | 5.89 |
| PPO: Taken classic boring suburban lifestyle and turned into boring 24yo. Looking for ideas for breaking out of rut and refreshing lifestyle. Any ideas welcome! | 5.91 |
| vBo$N$: boyfriend and I seem to have fallen into a rut of regularly doing mundane activities after working together non-stop for several months! ideas to spice things up? | 6.57 |
| Bo$N$: in a relationship that's getting a bit stale, looking for some inspiration to make changes on a whim in hopes of rejuvenating it! | 6.74 |

