# OpenReview forum: "Variational Best-of-N Alignment"
_ICLR.cc/2025/Conference — ICLR 2025 Poster_

### Official Review · Reviewer_Lgyc · 2024-11-01

**Soundness:** 3
**Presentation:** 3
**Contribution:** 3
**Rating:** 8
**Confidence:** 3

**Summary:**

This paper proposes to "distill" the gains from the best-of-N (BoN) inference-time alignment algorithm during finetuning, thereby improving a LLM's alignment without incurring the (linear in N) inference-time costs of BoN. They do this by deriving the distribution induced by the BoN algorithm, then defining the finetuning objective (which they call vBoN) to minimize the model's reverse KL divergence to this distribution. On the theoretical front, they derive some lower bounds for this vBoN objective and show that these lower bounds resemble the standard KL-constrained RL objective. Through experiments on controlled sentiment generation and summarization, they empirically show that PPO using the vBoN objective is the most effective technique for alignment via finetuning (though inference-time BoN still outperforms all finetuning-time alignment methods).

**Strengths:**

1. Clear motivation: vBoN is naturally motivated from the goal of reaping the gains of BoN without incurring the inference-time computational overhead.

2. Strong theoretical grounding, and compelling comparison of vBoN lower bounds with the standard KL-regularized RL objective.

3. Sound experimental setup: The win rate/average rewards vs KL curves in Figure 2 show a clean comparison against several important baselines, and the breakdown of how often each method appears on the Pareto front was also a nice statistic to report. The ablation and analysis to understand sensitivity to number of samples for the logF() approximation in the vBoN objective (Figure 3 and Table 1) were also clean and well-presented.

**Weaknesses:**

1. It is stated in the paper that, under some basic simplifying assumptions, the optimal distribution under the KL-regularized RL objective and the vBoN objective are asymptotically equivalent (Section 1, equation (2), Section 4). The paper also shows that lower bounds for the vBoN objective closely resemble the KL-constrained RL objective, and that models finetuned to maximize these lower bounds perform very similarly to those finetuned to maximize the vBoN objective. On the other hand, the paper shows that models finetuned to maximize the vBoN objective substantially outperform models finetuned to optimize the KL-constrained RL objective. This empirical evidence in some sense challenges the theoretical observations. Why, then (and under what conditions), is vBoN better than KL-constrained RL?

2. The paper demonstrates the effectiveness of vBoN on two tasks: Sentiment control (which is somewhat of a toy task) and Summarization (which is less of a toy task). However, several important baselines (e.g., DPO and BoNBoN) are only reported for the sentiment control task, and not for the summarization task. Why weren't these baselines included for the (less toy) summarization task?

See additional questions in the "Questions" section.

**Questions:**

1. The sentence right below equation (6) (the vBoN objective) states that "This is an entropy-regularized objective, where we...*discourage* the model from having low entropy". And this can be seen clearly in the objective (to be maximized) as well. However, in the text below equation (9), which is a lower bound for the vBoN objective, it is mentioned that L1(\theta) further *encourages* the model to have low entropy. Can you provide an interpretation of why this L1 lower bound encourages low entropy, while the original vBoN objective encourages high entropy?

2. Legend labels, as well as axis labels and numbers, are missing from Figure 4, which is one of the more important figures in the paper. Legend labels are also missing from Figures 5 and 6.

3. The abstract mentions that finetuning with the vBoN objective is "analogous to mean-field variational inference", but this parallel is not discussed in the main text of the paper.

4. In Section 5, the BoN-SFT baseline is considered, but its performance is not competitive since it has high KL divergence from the reference model. Was a KL-constrained version of BoN-SFT considered?

5. Like vBoN, both the the BoNBoN and BOND (Sessa et al, 2024) methods attempt to convert BoN to a finetuning-time alignment method. Why was BoNBoN, but not BOND, reported as a baseline in Section 5?

Nit: There is a typo at the end of Section 5, where the references to Fig. 6a and Fig. 6b should instead refer to Fig. 2a and Fig 2b.

---

> ### Author Response · Authors · 2024-11-21
> **Response to the reviewer**
>
> > This empirical evidence in some sense challenges the theoretical observations. Why, then (and under what conditions), is vBoN better than KL-constrained RL?
>
> Thank you for this thoughtful question.
> Regarding the relationship between the optimal policy under the KL-constrained RL objective and the BoN distribution: the simplifying assumptions in Yang et al. (2024)—namely, that the language model is memoryless and the reward is linear—are quite unrealistic. Additionally, the two distributions are proven to be asymptotically equivalent only in sequence length. For real-world alignment tasks, as studied in our experiments, these assumptions do not hold, and the distributions are not equivalent.
>
> As for the similarity between the lower bound and the KL-constrained objective, while we agree with the reviewer that their structures are indeed similar, there is a fundamental difference in how rewards are treated. With the KL-constrained RL objective, the goal is to maximize the expected reward. In contrast, with the lower bound, the objective is to maximize the probability that a random sample from $\pi_{\text{ref}}$​ has a reward value lower than that of strings sampled from the aligned policy. This theoretical distinction is what leads to the substantial differences observed in the empirical results.
>
> > several important baselines (e.g., DPO and BoNBoN) are only reported for the sentiment control task, and not for the summarization task. Why weren't these baselines included for the (less toy) summarization task?
>
> Thank you for your comment, we have added those baselines to the summarization experiments. In general, the conclusions still hold. We further observe that DPO and BoNBoN can only achieve competitive results in lower temperatures (0.25, 0.5) and their performance drops significantly at higher temperatures.
>
> **Questions**
>
> > The sentence right below equation (6) (the vBoN objective) states that "This is an entropy-regularized objective, where we...discourage the model from having low entropy". And this can be seen clearly in the objective (to be maximized) as well. However, in the text below equation (9), which is a lower bound for the vBoN objective, it is mentioned that L1(\theta) further encourages the model to have low entropy. Can you provide an interpretation of why this L1 lower bound encourages low entropy, while the original vBoN objective encourages high entropy?
>
> We agree with the reviewer that the terms in L1 might not be as intuitive as the original objective. The negative scalar for the entropy term arises from decomposing the objective into a KL divergence term and an entropy term. In L1​, the KL term prevents the model from diverging too far from the reference policy, while the entropy term encourages the exploitation of high-reward regions, creating a trade-off.
>
> > Legend labels, as well as axis labels and numbers, are missing from Figure 4, which is one of the more important figures in the paper. Legend labels are also missing from Figures 5 and 6.
>
> Thank you for pointing this out, though we’re unsure why this issue is occurring. The axis labels, numbers, and legends for Figures 4, 5, and 6 are visible on our end. Could this be related to the specific PDF viewer you are using?
>
> > The abstract mentions that finetuning with the vBoN objective is "analogous to mean-field variational inference", but this parallel is not discussed in the main text of the paper.
>
> Thank you for your feedback. We have incorporated a sentence in Section 3 to clarify the parallel to mean-field variational inference.
>
> > In Section 5, the BoN-SFT baseline is considered, but its performance is not competitive since it has high KL divergence from the reference model. Was a KL-constrained version of BoN-SFT considered?
>
> We did not consider adding a backward KL regularization term to SFT. However, it is worth noting that the reported BoN-SFT is equivalent to minimizing the forward KL divergence between the BoN distribution and the language model. That said, optimizing for both forward and backward divergences (as done in BOND) could potentially improve performance, albeit with increased computational costs.
>
> > Like vBoN, both the the BoNBoN and BOND (Sessa et al, 2024) methods attempt to convert BoN to a finetuning-time alignment method. Why was BoNBoN, but not BOND, reported as a baseline in Section 5?
>
> We agree that the BOND paper is relevant to our work. The primary reason we included BoNBoN as a baseline but not BOND is timing. BoNBoN was published a few weeks before we published our first draft, allowing us enough time to implement their method. In contrast, BOND was released after we had published our draft, leaving insufficient time to implement all the technical details, especially since the code for BOND has not yet been made available.

---

> > ### Comment · Reviewer_Lgyc · 2024-11-22
> >
> > Thank you for the clarifications, and for adding the DPO and BoNBoN baselines to the summarization experiments.

---

### Official Review · Reviewer_N6hp · 2024-11-01

**Soundness:** 2
**Presentation:** 2
**Contribution:** 2
**Rating:** 3
**Confidence:** 4

**Summary:**

This paper proposes a more direct preference-align method vBoN for LLMs based on the Best-of-N approach. The authors construct a new fine-tuning objective function using the scoring results from a reward model. By directly aligning the output distribution of the LLMs to the output distribution of the Best-of-N method, the authors aim to simplify the BoN inference process and achieve better alignment efficiency compared to other preference alignment methods. Experiment results are demonstrated to try to validate the claims.

**Strengths:**

The motivation of this article is quite clear, and it includes several experiments to support its claims. Additionally, the overall structure of the article is fairly complete, including some theoretical derivations.

**Weaknesses:**

1. The effectiveness of the vBon method is questionable. Although vBon utilizes reward model scoring to depict a target distribution closer to BoN, it requires a large number of samples (controlled by N or M in the article) to generate the corresponding preference data. The efficiency of this method is not high when the sample size is large.
2. The paper is hard to follow. Some definitions lack explanation and need clarification from the authors, such as the function F(.) used in Equations 4 and 5.
3. There is a lack of more convincing experiments to show the significant enhancement of alignment efficiency.

**Questions:**

1.Refer to the weakness 1 and 3, would you please further explain the efficiency of the method?
2.Refer to the weakness 2, would you please clarify the definition of your objective?

---

> ### Author Response · Authors · 2024-11-21
> **Response to the reviewer**
>
> > The effectiveness of the vBon method is questionable. Although vBon utilizes reward model scoring to depict a target distribution closer to BoN, it requires a large number of samples (controlled by N or M in the article) to generate the corresponding preference data. The efficiency of this method is not high when the sample size is large. There is a lack of more convincing experiments to show the significant enhancement of alignment efficiency.
>
> Thank you for raising this important question. We have added a subsection (5.2) devoted to analyzing the efficiency of vBoN. To summarize, there are 3 aspects affecting the efficiency of our method compared to other alignment methods. (i) vBoN is N times more efficient than BoN at inference time. This can be a huge improvement since BoN is normally used with large N values. (ii) N in vBoN acts as a regularization parameter, and **does not increase the optimization time**. Since we precompute the $F$ values, vBoN’s optimization loop runs as fast as PPO. (iii) The main computational overhead of vBoN (compared to PPO) is in the preprocessing step when we precompute $F$. In Figure 4 in the revised draft, we look at the performance vs. total elapsed time by increasing M. We observe that with as few as $M = 32$, which only took 10 minutes, we manage to match the performance of larger sample sizes.  We hypothesize that the data efficiency of the simple Monte Carlo estimator can be greatly improved by taking into account the similarity between different prompts to learn an approximation to $\log F$ function, which we plan as future work.
>
> > The paper is hard to follow. Some definitions lack explanation and need clarification from the authors, such as the function F(.) used in Equations 4 and 5.
>
> We appreciate your feedback and have worked to improve the paper's clarity. Specifically, we revised Sections 2 and 3 and adjusted the notation for better readability. Additionally, we now provide a more detailed explanation of the function $F$:
> “$F$ can be understood as the strict cumulative density function of reward values under $\pi_{\text{ref}}$. In other words, $F(r(y))$ represents the probability that a random sample drawn from $\pi_{\text{ref}}$ has a reward value less than $r(y)$.”
>
> We are committed to further improving the paper's clarity, so if there are additional areas that require explanation, please do not hesitate to let us know.

---

### Official Review · Reviewer_jVo6 · 2024-11-02

**Soundness:** 3
**Presentation:** 2
**Contribution:** 3
**Rating:** 6
**Confidence:** 4

**Summary:**

In this paper, the authors propose a novel optimization objective for reinforce ment learning in large language models: vBo𝑁. Inspired by the Bo𝑁 method during the inference phase, the authors have designed an innovative loss to aid model alignment training. Building on this, the authors have transformed the proposed loss into the easily optimizable vBo𝑁 by relaxing the optimization lower bound. The authors validated the potential of vBo𝑁 optimization on two tasks using the IMDB movie review dataset and the Reddit TLDR dataset, and compared it with other contemporary methods such as PPO and DPO, demonstrating the unique potential of the vBo𝑁 method.

**Strengths:**

- The authors conducted rigorous and thorough theoretical derivations in the paper, clarifying the process from the motivation behind the vBo𝑁 proposal to its transformation into an optimizable objective. This is highly beneficial for readers interested in optimization theory and can provide new insights for tackling more complex optimization problems.
- The vBo𝑁 method is highly effective. Moreover, despite the substantial theoretical derivations, the illustrations used by the authors to present results and compare methods are clear and easy to read, helping readers to quickly grasp the potential of vBo𝑁 for their applications.

**Weaknesses:**

- Less Persuasive Experiments: While we understand that conducting RLHF is always exceedingly costly, for instance, PPO requires maintaining four sets of model parameters, the fact that the validation of the vBo𝑁 method was only focused on movie review completion and text summarization datasets makes it lacks persuasiveness. We would like to understand the potential applications of vBo𝑁 in broader and more challenging tasks, such as code generation, mathematical problem solving, and multi-step reasoning. The experimental section of the paper is not as solid as the theoretical section.
- Lacking Efficiency Testing: One of the motivations behind the vBo𝑁 method is to address the additional overhead of the Bo𝑁 method during inference, which vBo𝑁 resolves by introducing additional training. Therefore, I believe that an analysis of the training costs of vBo𝑁 is a crucial issue that must be discussed in the paper. Based on the experimental results, I could even accept that the training efficiency of vBo𝑁 is slightly inferior to methods like PPO. Moreover, beyond absolute costs, relative costs are also worth considering, such as a comparison of the performance of vBo𝑁 and PPO under the same computational-power/time/data con straints. If the authors could address either of these two aspects, or preferably both, it would significantly enhance the completeness and the soundness of the paper.
- Several key symbols and terms in the formulas are not defined, making it challenging for readers to understand the equations' meaning. A glossary would greatly improve readability.
- In the "Summarization" section, the authors refer to the reward models with different names ("pythia-2.8B reward model" and "pythia-6.9B model"), but it’s unclear if these models differ in architecture or purpose (see Section 6).
- The experiments focus solely on summaries from the “relationship” and “relationship advice” subreddits for training and then evaluate performance on in-distribution and out-of-distribution Reddit posts.
- This paper provides inadequate detail when introducing the advantages and disadvantages of other alignment methods (such as RLHF and direct preference optimization). In particular, it lacks sufficient background information and details on how these methods are applied to text generation or summarization tasks , which may make it difficult for readers to understand the relative advantages of the methods proposed in this paper.
﻿
Others
-  In the paper, the authors refer to the input of the LLM as a ‘string’. Switching to the term ‘token’ would align better with most readers’ pref erence and expectations, especially there is no cross-tokenizer-aspect that requires natural language level operations in vBo𝑁.
- For those practitioners who are more focused on practical implementa tion, including a pseudocode for the vBo𝑁 alignment would enhance the readability of the paper. Moreover, vBo𝑁 does include more complex op erations beside the usual training pattern. I’d like to suggest authors to include one in the paper, even in the appendix.

**Questions:**

- "It is invariant to applying any monotonically increasing function to rewards" might not be entirely correct. Although monotonicity and insensitivity to reward scale are indeed related, the exact nature of this relationship is still unknown.
- The extent of the loss introduced by using the Monte Carlo estimator to maximize the lower bound of Equation (6) compared to the original formula remains unclear.
- The models used in the experiments by the authors are relatively outdated. I 'm curious if this method could achieve similar results on more recent models, such as the LLaMA-3 series.
- The trend of vBoN changes relatively smoothly with temperature adjustments; what causes this smoother trend?

---

> ### Author Response · Authors · 2024-11-21
> **Response to the reviewer (part 1)**
>
> >  Less Persuasive Experiments
>
> We appreciate your acknowledgment of the substantial costs of conducting experiments on broader tasks. The choice of experiments in our paper was informed by a review of similar work, including DPO, BoNBoN, and BOND. *Collectively*, these papers evaluate their methods on three main tasks, two of which we have included in this paper. We are also actively assessing the feasibility of adding the third task, Anthropic’s Helpful and Harmless benchmark, as part of our ongoing work.
>
> While we agree that extending vBoN to reasoning tasks would add significant value, our computational resources are currently limited. Nevertheless, we view this as a promising direction for future research and hope that our current results provide a solid foundation for further exploration of vBoN’s potential.
>
> > Lacking Efficiency Testing
>
> Thank you for this constructive feedback. We have added a subsection (5.2) devoted to analyzing the efficiency of vBoN. To summarize: (i) N in vBoN acts as a regularization parameter, and **does not increase the optimization time**. Since we precompute the $F$ values, vBo’s optimization loop runs as fast as PPO. (ii) The main computational overhead of vBoN (compared to PPO) is in the preprocessing step when we precompute $F$. In Figure 4 in the revised draft, we look at the performance vs. total elapsed time by increasing M. We observe that with as few as $M = 32$, which only took 10 minutes on a single A100 GPU, we manage to match the performance of larger sample sizes.  We hypothesize that the data efficiency of the simple Monte Carlo estimator can be greatly improved by taking into account the similarity between different prompts to learn an approximation to $\log F$ function, which we plan as future work.
>
> > Several key symbols and terms in the formulas are not defined, making it challenging for readers to understand the equations' meaning. A glossary would greatly improve readability.
>
> We have made significant improvements to the clarity of the paper. Specifically, we added the missing explanation of notation in Sections 2 and 3 and included a notation glossary in the appendix.
>
> > In the "Summarization" section, the authors refer to the reward models with different names ("pythia-2.8B reward model" and "pythia-6.9B model"), but it’s unclear if these models differ in architecture or purpose (see Section 6).
>
> The smaller model is used within the optimization loop of alignment methods, such as in the PPO fine-tuning process to compute reward values. However, to ensure the robustness of our empirical evaluation against reward hacking, we use a separate, larger reward model exclusively for evaluation. This distinction has been clarified in the revised draft.
>
> > The experiments focus solely on summaries from the “relationship” and “relationship advice” subreddits for training and then evaluate performance on in-distribution and out-of-distribution Reddit posts.
>
> `relationship` and `relationship advice` are the two big categories in the TL;DR dataset, and together they cover 50% of the dataset. We chose this splitting so that we have a large portion of the dataset that we can meaningfully learn from, but also a held-out dataset with different topics that enable us to test out-of-distribution performance.
>
> > This paper provides inadequate detail when introducing the advantages and disadvantages of other alignment methods (such as RLHF and direct preference optimization). In particular, it lacks sufficient background information and details on how these methods are applied to text generation or summarization tasks , which may make it difficult for readers to understand the relative advantages of the methods proposed in this paper.
>
> Thank you for your thoughtful feedback. Section 2 is intended to provide all the necessary background knowledge for understanding prior work and distinguishing vBoN from these methods. However, your comment prompted us to revisit Section 2, and we realized that we had not fully explained how a reward model is constructed for open-ended text generation tasks, such as summarization. To address this, we added clarification in footnote 4:
>
> “For example, in a summarization task, a preference dataset consists of a document, two candidate summaries for that document, and a label indicating which summary is preferred by humans. The reward model is trained on this dataset to maximize the likelihood of correctly predicting human preference.”
>
> We hope this addition provides the necessary context and improves the clarity of our discussion.

---

> > ### Author Response · Authors · 2024-11-21
> > **Response to the reviewer (part 2)**
> >
> > > For those practitioners who are more focused on practical implementation, including a pseudocode for the vBo𝑁 alignment would enhance the readability of the paper.
> >
> > Thank you for this suggestion. We added a pseudocode in Appendix D.
> >
> > > "It is invariant to applying any monotonically increasing function to rewards" might not be entirely correct. Although monotonicity and insensitivity to reward scale are indeed related, the exact nature of this relationship is still unknown.
> >
> > That is true, we should have said “strictly” monotonically increasing function. This is adjusted in the revised draft. To answer your question regarding the relationship between the invariance property and scales, we should note that the invariance implies the insensitivity. Formally, the invariance property means that $F(R) = F(g(R))$ where $g$ is a strictly monotonically increasing function: $g: \mathbb{R} \rightarrow \mathbb{R}, r(y) < r(y’) \Leftrightarrow g(r(y)) < g(r(y’))$. This implies that $F$ is also invariant to reward scales. This is because we can set $g$ as a scaling function, i.e., $g(\cdot)$ = positive scaler $\times r(\cdot)$ + bias.
> >
> > > The extent of the loss introduced by using the Monte Carlo estimator to maximize the lower bound of Equation (6) compared to the original formula remains unclear.
> >
> > This question touches on two key aspects. The first concerns the accuracy of the Monte Carlo estimator, and the second addresses the effectiveness of maximizing the lower bound instead of the original objective.
> >
> > We address the first aspect in Section 5.1, where we demonstrate that the Monte Carlo estimator converges to the exact value with as few as 200 samples. For the second aspect, we provide evidence in Appendix E showing that maximizing the lower bound yields results very close to those obtained by directly maximizing the objective.
> >
> > > The models used in the experiments by the authors are relatively outdated. I 'm curious if this method could achieve similar results on more recent models, such as the LLaMA-3 series.
> >
> > In our experiments, we used Pythia models of comparable size, which are fully open and provide complete transparency in training. Unlike LLaMA-3 Instruct models, Pythia models have not undergone any preference optimization steps, avoiding potential confounding factors in our experiments. We agree with the reviewer that scaling vBoN to larger and more diverse models would provide additional value. However, due to resource constraints, we focused on the best setup we could achieve within our limitations.
> >
> > > The trend of vBoN changes relatively smoothly with temperature adjustments; what causes this smoother trend?
> >
> > Thank you for reading the paper closely and raising this insightful question. While we cannot be certain, we speculate that the smoother trend of vBoN with temperature adjustments could be due to vBoN’s reliance on ranking comparisons rather than absolute reward values, but we acknowledge that further investigation is needed to confirm this hypothesis.

---

### Official Review · Reviewer_DQFD · 2024-11-12

**Soundness:** 3
**Presentation:** 2
**Contribution:** 3
**Rating:** 6
**Confidence:** 3

**Summary:**

The best-of-N alignment strategy has proven very useful in generating text with high-rewards that still has a high probability under the reference model. Several studies have shown that BoN often outperforms models simply fine-tuned with RLHF. However, this improvement in BoN comes at computational overhead during inference.

This paper proposes Variational BoN (vBoN), a scheme that converts BoN from an alignment-via-inference algorithm to an alignment-via-fine-tuning algorithm. Basically, vBoN  derives the probability distribution induced by the BoN algorithm and then approximates this distribution by minimizing the reverse KL divergence between the language model and the BoN distribution. The model is optimized for the vBoN objective using the PPO algorithm.

The proposed objective is evaluated on controlled generation and summarization tasks, showing performance close to that of the BoN algorithm while being as cost effective as inference on the original reference model.

**Strengths:**

vBoN is a novel and effective approach converting BoN from an alignment-via-inference algorithm to an alignment-via-fine-tuning algorithm. Models fine tuned with the vBoN objective achieves high reward values closer to the BoN approach, while achieving probabilities closer to the reference model. Importantly, it is as cost-effective as inference on the original reference model. In comparison , the original BoN approach is N times more expensive.

Provided theoretical connections showing how vBoN is compared with other alignment objectives

**Weaknesses:**

Section 2 and 3 can be improved significantly by improving the notations and explanations and by bringing important details from Appendix to the main part of the paper. Currently I often find these two sections a bit confusing as well as a bit hard to appreciate some of the claims the authors have made in the paper. For example, in Eq 4, F(r(y)) will be defined as F(r(y) ) = P (r(y) < r(y) ), using Eq 5?  With this I am not sure how the vBoN objective is insensitive to applying any monotonically increasing function to the reward values?

Another important weakness of the paper is the current evals in the paper. Despite using the standard benchmarks for controlled generation and summarization, the evaluation chose not to report on standard metrics along with rewards and proximity to reference model comparisons.

**Questions:**

Please see my comments in the Weakness part of the reviews.

Why BoNBoN is called with that name? Please explain.

“We visualize the win rate vs. KL curves in Fig. 6a, and Fig. 6b the average rewards of generations under πθ vs. the KL divergence”. Please mention that these figures are in the appendix.

---

> ### Author Response · Authors · 2024-11-21
> **Response to the reviewer**
>
> Thank you for reading our  work and providing feedback.
> > Section 2 and 3 can be improved significantly by improving the notations and explanations
>
> Thank you for your constructive feedback. We have made significant improvements to the clarity of Sections 2 and 3. Specifically, we added the missing explanation of notation in Section 2 and included a notation glossary in the appendix. Additionally, we expanded the definition of the $F$ function in section 3 and provided further elaboration on the monotonic invariance property. We hope these changes address your comment, and we welcome any further suggestions for improving clarity in these sections.
>
> > For example, in Eq 4, F(r(y)) will be defined as F(r(y) ) = P (r(y) < r(y) ), using Eq 5? With this I am not sure how the vBoN objective is insensitive to applying any monotonically increasing function to the reward values?
>
> We noticed that the use of $R$ in Equation 5 caused some confusion, particularly in comparison to Equation 4. To address this, we have revised the notation for greater clarity, and it now reads as $F(r(y)) = \underset{y' \sim \pi_{\text{ref}}}{\mathbb{P}} (r(y’) < r(y))$.
>
> Regarding your question, the function $F$ depends solely on the comparison between two reward values—for example, whether the reward for a string $r(y)$ is greater than that for another string $r(y’)$. As a result, applying a strictly monotonically increasing transformation to all reward values will not affect $F$.
>
> > the evaluation chose not to report on standard metrics along with rewards and proximity to reference model comparisons.
>
> Thank you for your comment, though we are unsure if we fully understand your concern. We reported two standard metrics: average reward values and win rates, along with proximity to the reference model, measured using KL divergence. Could you clarify which additional metrics you would like to see?
>
> > Why BoNBoN is called with that name? Please explain.
>
> We did not change the name of the algorithm and used the name [the authors](https://arxiv.org/pdf/2406.00832) have chosen for their method.
>
> > “We visualize the win rate vs. KL curves in Fig. 6a, and Fig. 6b the average rewards of generations under πθ vs. the KL divergence”. Please mention that these figures are in the appendix.
>
> This was, in fact, a typo and is fixed now. We meant to refer to the figures in the main body of the paper. Thank you for catching this.

---

> > ### Comment · Reviewer_DQFD · 2024-11-26
> > **standard metrics**
> >
> > Sorry for the late response. And thanks for addressing my comments and improving the paper accordingly.
> >
> > - the evaluation chose not to report on standard metrics along with rewards and proximity to reference model comparisons.
> > - Could you clarify which additional metrics you would like to see?
> >
> > For controlled generation and summarization, it would have been interesting to report performance on Rouge or Blue capturing fluency and metrics for factuality or consistency.

---

> > > ### Author Response · Authors · 2024-11-26
> > >
> > > Thank you for the suggestion. For the controlled generation experiment, there are no reference outputs, making metrics like ROUGE or BLEU inapplicable. Regarding summarization, we note that it is standard in the RLHF and preference optimization literature not to report automatic metrics like ROUGE. This is because reward models trained on preference data have been shown to correlate significantly better with human judgments than automatic heuristic metrics such as ROUGE. For a detailed discussion on this, please refer to Appendix G.7 of the [Learning to Summarize from human feedback paper](https://arxiv.org/pdf/2009.01325).
> > >
> > > That said, we understand the value of including these metrics for completeness and broader comparability. We will consider adding them to the summarization experiments in the final version of the paper.

---

### Author Response · Authors · 2024-11-21
**General Response to Reviewers**

We thank all reviewers for their thoughtful and constructive feedback, which has helped us refine and clarify the paper. We are pleased that reviewers found our approach novel, well-motivated, and grounded in rigorous theoretical foundations, with clear derivations and insightful experimental comparisons.

In response to the feedback, we have made targeted revisions to improve the clarity and completeness of the paper while maintaining its original focus. The revised draft includes the following updates, with changes highlighted in green:

1. Efficiency analysis: We added a detailed discussion of vBoN’s efficiency compared to BoN and PPO, covering inference, optimization, and preprocessing costs.
2. Improved clarity in section 2: To enhance readability, we expanded the background on RLHF, provided clearer explanations of notation, and included a glossary of notation in the appendix.
3. Clarifications in section 3: Several reviewers requested more detail on the $F$ function. We added a comprehensive explanation and adjusted the notation to improve readability.
4. Adding the missing baselines: We incorporated BoNBoN and DPO baselines in the summarization experiments to provide a more comprehensive evaluation.

---

### Meta-Review · Area_Chair_soqk · 2024-12-15

**Metareview:**

Best-of-N (BoN) aligns language models to human preferences by selecting the highest-reward sample from N outputs, but it is computationally expensive.  Variational BoN (vBoN) addresses this limitation by fine-tuning the model to approximate BoN’s distribution, thereby reducing inference costs while maintaining strong performance.

This paper is well-written and easy to understand.  Compared to BoN, vBoN demonstrates significantly higher inference efficiency.  The authors have incorporated reviewers' suggestions, including adding an efficiency analysis, improving clarity, and providing additional baselines.  These revisions enhance the quality of the work, and I believe the paper merits acceptance.

I also recommend that any additional experiments promised by the authors in their response to reviewers are thoroughly completed and included in the final version.

**Additional Comments On Reviewer Discussion:**

Reviewers DQFD, jVo6, and N6hp highlighted that the paper lacks detailed evaluations and suggested improving the notations and explanations in the main sections. The authors have made the necessary revisions to address these concerns.

Reviewer N6hp also questioned the efficiency of the proposed method and noted the lack of detailed experiments. The authors acknowledged these limitations and left them to future work.

Similarly, Reviewer Lgyc pointed out the absence of detailed experimental results. The authors addressed his/her concerns.

In summary, most of the feedback centered on improving the paper's writing quality and the lack of comprehensive experiments. The authors effectively addressed most of the issues during the rebuttal process. Therefore, I recommend accepting the paper.

---

### Decision · Program_Chairs · 2025-01-22

Accept (Poster)